# SALIENCE AWARE MARK-STEERED PROMPTING FOR LLMS

## ABSTRACT

For the same task, more detailed context provided in an LLM prompt usually leads to better performance. However, large-size prompts can cause LLMs to lose focus on the most critical instructions, a phenomenon known as *attention dilution*, which can significantly degrade model performance on complex tasks requiring long prompts. To address this issue, we propose *Salience Aware Mark-Steered Prompting* (**MSP**), a novel framework designed to mitigate attention dilution by explicitly steering the model's focus toward the most critical information within a prompt. MSP consists of two stages: first, Gradient-Guided Mask Search (GGMS) automatically identifies the most influential tokens. Second, Mark-Steered Decoding (MSD) persistently guides the model by amplifying the influence of these key tokens at each step of the inference process, improving the model's alignment with core user instructions. We evaluated the effectiveness of MSP on five widely used benchmarks and three representative LLMs with varying scales. The experimental results demonstrate that MSP exhibits consistent performance gains over state-of-the-art baselines, and its strong performance across diverse tasks and models highlights its robustness and generalizability. Our implementation code is provided in the supplementary materials.

## 1 INTRODUCTION

Large language models (LLMs) have demonstrated strong performance across a variety of tasks (Colombo et al., 2024; Zhang et al., 2024b; Qin et al., 2023a; Touvron et al., 2023), and prompting has become the primary interface through which users interact with these models (Liu et al., 2023b). In recent years, a large number of prompt optimization methods have been proposed to further unlock the potential of LLMs (Pryzant et al., 2023; Lin et al., 2024; Chen et al., 2024a; Wang et al., 2024; Ye et al., 2024; Hu et al., 2024). Most of these approaches rely on evolutionary algorithms that iteratively modify existing prompts (Fernando et al., 2024; Guo et al., 2024; Agrawal et al., 2025), or on meta-prompts that incorporate scoring mechanisms over a small validation set to refine prompts (Zhou et al., 2023; Yang et al., 2024; Wu et al., 2024). Collectively, such methods are often referred to as prompt engineering. More recently, the concept of context engineering has been introduced (Mei et al., 2025). Unlike traditional prompt engineering, context engineering focuses on providing LLMs with complete information needed to perform subsequent reasoning or generation tasks. Beyond conventional prompts, it often includes additional information obtained from retrieval-augmented generation (RAG) (Gao et al., 2023) and tool calling (Shen, 2024).

However, more context information does not necessarily lead to better performance. A phenomenon known as *attention dilution* (Tian & Zhang, 2025) has been observed and empirically verified (Qin et al., 2022; Zhang et al., 2024c; Fang et al., 2025; Qin et al., 2023b), which shows that LLMs may fail to identify and focus on the truly important parts of the input (Liu et al., 2023a). Specifically, as the LLM proceeds with autoregressive generation, it tends to forget the salient information contained in the input, which may result in degraded performance. This issue can be better understood through an analogy with human reading comprehension. When humans read a long document, we often highlight or underline important words or sentences to quickly grasp the main ideas and better understand the content. Inspired by this intuition, this motivates us to investigate whether a similar strategy can be applied to LLMs: *if the salient parts of a prompt can be highlighted, can we enable the LLM to maintain focus on those parts throughout the inference process, thereby unlocking its full potential?*

To this end, we propose *Salience Aware Mark-Steered Prompting*(**MSP**), a two-stage framework. In the first stage, we assign a set of masks to the tokens of the input prompt and use a gradient-based search strategy to identify the optimal mask configuration. The optimization objective is to maximize the change in the model's output, which allows us to automatically detect the most salient tokens in the prompt. In the second stage, we ensure that the model consistently attends to these salient tokens at every decoding step. Specifically, we compute the difference in logits between the masked and unmasked versions of these salient tokens, derive representations from this difference, and then apply a simple linear amplification. These enhanced representations are added back to the model's original logits at each decoding step. Through these two stages, MSP can automatically identify salient prompt content and leverage it to guide the generation process of LLMs.

In summary, our main contributions are as follows:

- Motivated by the human practice of highlighting key information in documents to aid comprehension, we propose MSP, which automatically identifies salient prompt content and leverages it to guide LLM generation.

- We develop a gradient-based heuristic mask search algorithm that efficiently identifies salient tokens in prompts, while significantly reducing computational overhead.

- We introduce a simple yet effective method for sustaining LLM focus, which reinforces attention to salient content through lightweight linear operations, thereby guiding the generation process.

- We conduct extensive experiments across a diverse set of natural language processing tasks to demonstrate the effectiveness and scalability of the proposed framework.

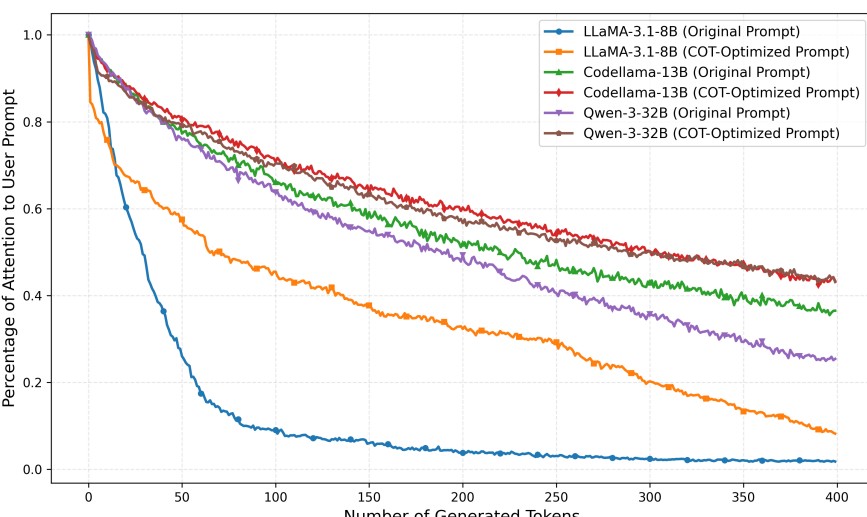

Figure 1: Attention percentage allocated to the user prompt across the generation process for three LLMs under two prompting conditions (Original vs. COT-Optimized).

## 2 EMPIRICAL STUDY OF ATTENTION DILUTION

To systematically examine the phenomenon of *attention dilution*, we conduct an empirical study on three representative open-source LLMs: LLaMA-3.1-8B, CodeLlama-13B, and Qwen-3-32B, using both the *original prompt* and a *COT-Optimized prompt*. Following prior work (Selvaraju et al., 2016), we compute the percentage of attention allocated to the user prompt during autoregressive generation via a gradient-based method (See Details in Appendix A.1). Across all models on HumanEval, we observe clear attention drift: user prompt attention consistently decreases as generation proceeds, with LLaMA-3.1-8B showing the most severe drop, falling below 10% of its initial level (Figure 1), which aligns with its lowest task performance. While the COT-Optimized prompt slows the decay and maintains higher attention for longer, it does not prevent the eventual

decline, demonstrating that attention dilution is a persistent and fundamental issue that arises across all model inputs. This limitation highlights a key bottleneck in long-context reasoning and motivates our method aimed at mitigating such attention-shift behavior.

# 3 PRELIMINARIES

Given a user prompt $x$ and an LLM $f_\theta$, the model first tokenizes it into a sequence of tokens $\boldsymbol{x} = (x_1, x_2, \ldots, x_n)$, where $x_i \in \{1, 2, \ldots, |V|\}$, $|V|$ denotes the vocabulary size, and $n$ is the length of the input sequence. The set of token indices is $\mathcal{I} = \{1, 2, \ldots, n\}$. The corresponding generated output $\boldsymbol{y}$ is represented as a sequence of tokens $\boldsymbol{y} = (t_1, \ldots, t_l)$ with $t_j \in \{1, 2, \ldots, |V|\}$. The output tokens $\{t_j\}_{j=1}^l$ are generated autoregressively. Specifically, at step $i$, the input to $f_\theta$ is an $n \times d$ embedding matrix $\mathbf{E}_i$, defined as:

$$\mathbf{E}_i = \left[\mathbf{E}^x, \mathbf{t}_1, \mathbf{t}_2, \ldots, \mathbf{t}_{i-1}, \texttt{PAD}\right], \tag{1}$$

where $\mathbf{E}^x$ is the submatrix of embeddings corresponding to the tokens in the user prompt $\boldsymbol{x}$, $\mathbf{t}_1, \ldots, \mathbf{t}_{i-1}$ are the embeddings of previously generated tokens, and $\texttt{PAD}$ denotes a padding submatrix. The model outputs logits, which are converted into a probability distribution via the softmax function. The next token $t_i$ is then selected using a sampling strategy, formally expressed as:

$$t_i = \arg\max_t \mathrm{softmax}(f_\theta(\mathbf{E}_i)) = \arg\max_t P_\theta(t \mid \boldsymbol{x}, t_1, \ldots, t_{i-1}), \tag{2}$$

where $f_\theta$ denotes the model's output function parameterized by $\theta$. However, during autoregressive generation, the model inherently suffers from attention dilution. As the generation proceeds, the attention weights assigned to salient tokens in the prompt tend to diminish, dispersing instead across less relevant recent context. For example, in the code generation task with the instruction "Write a function that sorts the given list of integers in ascending order according to the sum of their digits", the phrase "in ascending order" carries particularly important information. Yet the model lacks the ability to automatically identify and prioritize such salient tokens. On the one hand, during the decoding stage the model has no knowledge of which tokens are more important, and on the other hand, it lacks a mechanism to increase its attention to them. These limitations restrict the model's overall performance.

# 4 METHOD

In this section, we introduce Salience Aware **M**ark-**S**teered **P**rompting (MSP), a simple yet effective two-stage framework. In the first stage, a gradient-based heuristic search strategy is employed to identify a subset of tokens in the input prompt that are both important and semantically related. In the second stage, at each decoding step of the LLM, we derive representations of these salient tokens and apply a lightweight linear amplification to enhance the model's original output logits, thereby improving performance across a wide range of tasks. The overall architecture of our framework is illustrated in Figure 2.

## 4.1 GRADIENT-GUIDED MASK SEARCH

Given a sequence of tokens $\boldsymbol{x} = (x_1, x_2, \ldots, x_n)$, our objective in this stage is to identify a subset of $k$ tokens that exert the greatest influence on the generated sequence. To formalize this, we introduce a binary prompt mask $\boldsymbol{m} = (m_1, \ldots, m_n)$, where $m_i \in \{0, 1\}$ indicates whether the $i$-th token is retained or masked. The joint probability of producing the original output sequence $\boldsymbol{y}$ given a masked input $\boldsymbol{m} \odot \boldsymbol{x}$ is defined as $p_\theta(\boldsymbol{y}|\boldsymbol{m} \odot \boldsymbol{x})$, where $\odot$ denotes the Hadamard product. The goal is to identify a binary mask $\boldsymbol{m}$ that maximizes the discrepancy between the probability of generating $\boldsymbol{y}$ from the full input $\boldsymbol{x}$ and that from the masked input $\boldsymbol{m} \odot \boldsymbol{x}$. A larger discrepancy indicates that the masked tokens correspond to the most influential components for producing $\boldsymbol{y}$ and should therefore be considered the important subset. This leads to the following optimization problem:

$$\max_{\boldsymbol{m} \in \{0,1\}^n} \mathcal{L}(\boldsymbol{m}, \boldsymbol{x}, \boldsymbol{y}; \boldsymbol{\theta}) = p_\theta(\boldsymbol{y}|\boldsymbol{x}) - p_\theta(\boldsymbol{y}|\boldsymbol{m} \odot \boldsymbol{x}) \tag{3}$$

where $k$ specifies the number of salient tokens to be identified. Intuitively, one major challenge in solving Eq. (3) lies in the enormous search space of binary masks, which grows rapidly with the input

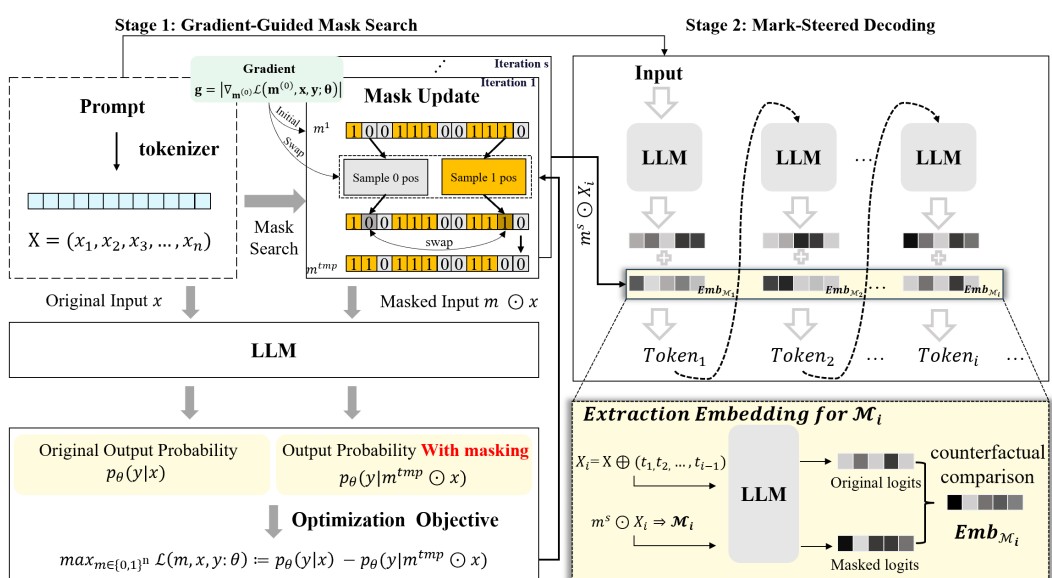

Figure 2: Overview of the proposed Salience Aware Mark-Steered Prompting (MSP). The framework consists of two stages: (i) Gradient-Guided Mask Search, which identifies salient tokens in the input prompt, and (ii) Mark-Steered Decoding, which reinforces the influence of these tokens at every decoding step.

length. For an input consisting of $n$ tokens, the task reduces to selecting $k$ tokens out of $n$, and the number of possible masks is given by the binomial coefficient $\binom{n}{k} = \frac{n!}{k!\,(n-k)!}$. The corresponding computational complexity is $\mathcal{O}(n^k)$. This combinatorial growth makes exhaustive search infeasible for typical prompt lengths. To address this problem, we propose **G**radient-**G**uided **M**ask **S**earch (**GGMS**), a heuristic strategy consisting of two components. First, we design a gradient-guided mask initialization method to provide a more effective starting point for the search. Second, we introduce a gradient-guided mask update strategy to improve the efficiency of the search process. It is worth noting that although both steps rely on gradients, GGMS requires computing gradients only once. We describe these two strategies in detail below.

**Mask Initialization.** To improve the efficiency of the search, we first focus on optimizing the starting point by constructing a more informative initialization mask. Gradients are widely recognized as reliable indicators of feature importance, as demonstrated in prior work (Sundararajan et al., 2017; Selvaraju et al., 2017; Kapishnikov et al., 2021). Motivated by this observation, we employ gradients to guide the initialization of the binary mask. We begin with an initial mask $\boldsymbol{m}^{(0)} = \mathbf{1}$, where all tokens in the input $\boldsymbol{x}$ are treated as non-influential. We then compute the gradient of the loss function in Eq. (3) with respect to this mask: $\nabla_{\boldsymbol{m}^{(0)}} \mathcal{L}(\boldsymbol{m}^{(0)}, \boldsymbol{x}, \boldsymbol{y}; \boldsymbol{\theta})$, and take the absolute values of its components, denoted as $\boldsymbol{g} = |\nabla_{\boldsymbol{m}^{(0)}} \mathcal{L}(\boldsymbol{m}^{(0)}, \boldsymbol{x}, \boldsymbol{y}; \boldsymbol{\theta})|$. Tokens associated with larger entries in $\boldsymbol{g}$ are expected to have a stronger influence on the generated output if altered. Based on this signal, we construct the first updated mask $\boldsymbol{m}^{(1)}$ by setting $m_i^{(1)} = 0$ whenever $g_i$ belongs to the top-$k$ values of $\boldsymbol{g}$. In this way, gradient information directly guides the identification of salient tokens for initialization, yielding a stronger starting point for subsequent optimization.

**Gradient-Guided Mask Update.** After obtaining the initial mask, the next step is to iteratively refine it according to the optimization objective in Eq. (3) in order to identify a better combination of tokens. We perform $s$ iterations of updates. At each iteration, given the current binary mask $\boldsymbol{m}$, we sample one non-zero position $p$ (corresponding to a non-influential token) and one zero position $q$ (corresponding to an influential token), and then swap their values to explore a new candidate mask. The sampling process is guided by the gradient information computed in the initialization stage. Specifically, non-zero positions are sampled according to probabilities derived from

the normalized gradient magnitudes, expressed as $\mathrm{softmax}(\boldsymbol{m}^{(s)} \odot \boldsymbol{g})$. A similar gradient-guided sampling strategy is applied to zero positions. Once the two positions $p$ and $q$ are selected and swapped, we obtain a temporary mask $\boldsymbol{m}^{\mathrm{tmp}}$. We then evaluate whether $\boldsymbol{m}^{\mathrm{tmp}}$ reduces the probability of generating the output sequence $\boldsymbol{y}$. If the probability decreases, the binary mask is updated to $\boldsymbol{m}^{\mathrm{tmp}}$; otherwise, the original mask is retained. After $s$ iterations of updates, we obtain an optimal mask, and the tokens corresponding to the zero positions constitute the ***Mark Text***, defined as $\mathcal{M} = (x_{i_1}, x_{i_2}, \ldots, x_{i_k})$, $\{i_1, i_2, \ldots, i_k\} \subseteq \{1, 2, \ldots, n\}$, where $\mathcal{M}$ represents the subset of salient tokens selected from the input sequence $\boldsymbol{x} = (x_1, x_2, \ldots, x_n)$.

This sampling strategy ensures that both non-influential and influential tokens are selected according to probabilities derived from their gradient magnitudes. As a result, tokens with larger gradients are more likely to be swapped, while tokens with smaller gradients still retain a non-negligible chance of being chosen. Such a design prevents the search from being overly greedy and enables the algorithm to explore a broader solution space. At the same time, it leverages gradient information to guide the optimization toward more promising regions. In this way, the strategy achieves a balance between exploration and exploitation, which is essential for effective search in the discrete mask space.

## 4.2 MARK-STEERED DECODING

After identifying salient texts through GGMS, we proceed to **M**ark-**S**teered **D**ecoding (**MSD**), where these *marked texts* $\mathcal{M}$ are leveraged to guide the generation process. Specifically, the model's attention and output distribution are subtly adjusted to reinforce the influence of salient texts, ensuring that key information identified in Stage one has a stronger impact on the final generated output.

The core motivation of this stage is as follows: if, at every decoding step, the model can leverage a real-time contextual representation of the *marked texts* $\mathcal{M}$, and this representation is incorporated as an additional signal to adjust the output distribution, then the model can be consistently guided to align with user intent. To achieve this goal, we introduce two key components: (i) the extraction of contextual representation for $\mathcal{M}$, and (ii) decoding guided by the augmented influence of $\mathcal{M}$.

**Extraction of Contextual Representation for $\mathcal{M}$.** We first aim to obtain a reliable representation that captures the influence of the *marked texts*. A desirable representation must satisfy two criteria: (1) it should be context-dependent, reflecting the role of the *marked texts* in the full input; and (2) it should directly intervene in the model's decision process at the logits level. Static representations, such as using token embeddings of the *marked text*, are insufficient because they cannot capture the complex semantic interactions with other context, including the already generated tokens. To address this limitation, we propose a dynamic extraction method based on counterfactual comparison. We first define the masked embedding matrix at step $i$ as:

$$\mathbf{E}_i^{\mathrm{mask}} = \left[ \mathbf{E}^{x \setminus \mathcal{M}}, \mathbf{t}_1, \mathbf{t}_2, \ldots, \mathbf{t}_{i-1}, \mathtt{PAD} \right], \tag{4}$$

where $\mathbf{E}^{x \setminus \mathcal{M}}$ denotes the embeddings of the user prompt $x$ with all tokens in $\mathcal{M}$ replaced by a special mask token. Let $\boldsymbol{F}_i$ denote the logits computed by the model $f_\theta$ at decoding step $i$. At this step, we compute the influence representation of $\mathcal{M}$, denoted as $\boldsymbol{v}_{\mathrm{influence}}$, by taking the difference between two distributions.

- **Original logits** ($\boldsymbol{F}_i^{\mathrm{original}}$): the logits obtained when the LLM processes the full input that contains $\mathcal{M}$,

$$\boldsymbol{F}_i^{\mathrm{original}} = f_\theta(\mathbf{E}_i). \tag{5}$$

- **Masked logits** ($\boldsymbol{F}_i^{\mathrm{masked}}$): the logits obtained when the tokens in $\mathcal{M}$ are replaced with a semantically empty mask,

$$\boldsymbol{F}_i^{\mathrm{masked}} = f_\theta(\mathbf{E}_i^{\mathrm{masked}}). \tag{6}$$

The contextual representation of $\mathcal{M}$ at decode step $i$ is then defined as:

$$\boldsymbol{v}_{\mathrm{influence}} = \boldsymbol{F}_i^{\mathrm{original}} - \boldsymbol{F}_i^{\mathrm{masked}}. \tag{7}$$

Each dimension of $\boldsymbol{v}_{\mathrm{influence}}$ quantifies how the presence of $\mathcal{M}$ shifts the model's predictive preference for the corresponding vocabulary item, thereby serving as a differential guidance signal. Additional theoretical analysis is given in Appendix A.2.

**Decoding Guided by the Augmented Influence of** $\mathcal{M}$**.** After obtaining the representation $\boldsymbol{v}_{\text{influence}}$, our objective is to amplify its effect at every decoding step so that the LLM maintains sustained attention to the *Mark Text*. To achieve this, we introduce a hyperparameter $\omega$, referred to as the *mark strength*, which controls the degree of amplification. The augmented logits at decoding step $i$ are computed as:

$$\boldsymbol{F}_i^{\text{augmented}} = \boldsymbol{F}_i^{\text{original}} + \omega \cdot \boldsymbol{v}_{\text{influence}} = f_\theta(\mathbf{E}_i) + \omega \cdot \big(f_\theta(\mathbf{E}_i) - f_\theta(\mathbf{E}_i^{\text{masked}})\big). \tag{8}$$

This formulation explicitly reinforces the contribution of the *Mark Text* at each decoding step, thereby ensuring that the model consistently attends to the salient instructions specified in the prompt. Moreover, the procedure is lightweight and requires no additional training, making it broadly applicable across different models and tasks.

## 5 EXPERIMENT SETUP

### 5.1 DATASETS AND BASELINES

We evaluate the effectiveness of MSP on five widely used benchmarks, covering diverse task types. Specifically, we consider two code generation benchmarks, HumanEval (Chen et al., 2021) and MBPP (Austin et al., 2021), two text generation benchmarks, TruthfulQA (Lin et al., 2021) and MMLU (Hendrycks et al., 2021), and one mathematical reasoning benchmark, GSM8K (Cobbe et al., 2021). See Appendix A.3 for more details on the datasets.

For base models, we select three representative open-source large language models ranging in scale from 8B to 32B parameters: LLaMA-3.1-8B (Dubey et al., 2024), CodeLlama-13B (Roziere et al., 2023), and Qwen-3-32B (Yang et al., 2025). We compare our method against four state-of-the-art baselines, including attention-steering approaches such as PASTA (Zhang et al., 2024a) and SPA (Tian & Zhang, 2025), as well as prompting-based methods such as Self-Debugging (Chen et al., 2024b) and Self-Planning (Jiang et al., 2024). The latter two prompting methods are evaluated only on code generation tasks. This setup ensures a comprehensive evaluation across both task types and model scales.

### 5.2 EVALUATION METRICS

To comprehensively assess MSP, we adopt the following evaluation metrics tailored to each task. For code generation tasks, we measure the model's ability to produce functionally correct code using the Pass@1 metric, which represents the proportion of problems for which at least one generated solution passes all unit tests. For the TruthfulQA dataset, following the established benchmark methodology, we employ the DeepSeek-V3 model (DeepSeek-AI, 2024) as a judge to evaluate the generated answers in terms of *truthfulness* and *informativeness*. For the MMLU and GSM8K datasets, we adopt *accuracy* as the primary metric to assess the model's multi-domain knowledge and mathematical reasoning abilities. More details are provided in Appendix A.4.

### 5.3 IMPLEMENTATION DETAILS

In the first stage of MSP, the objective is to identify the top $k$ tokens that most strongly influence the generated sequence, achieved through $s$ iterative updates with a masking procedure. In the second stage, we amplify the effect of these influential tokens by scaling their contribution with a hyperparameter $\omega$ at each decoding step. Thus, MSP involves three key hyperparameters. We set $k$ dynamically as half of the input prompt length, $s$ to 5 considering computational efficiency, and determine $\omega$ through grid search on the dataset to select the value that yields the best performance across tasks. More details are provided in Section 6.4 and Appendices A.5 and A.9.

## 6 RESULTS

### 6.1 IMPROVEMENT OVER BASE MODELS

As shown in Table 1, MSP achieves consistent and significant performance improvements across three representative open-source large language models on five diverse benchmarks. These gains

are especially pronounced on tasks that require generating long-form content, such as code generation and open-ended question answering. For example, on HumanEval our method brings a substantial 12.2-point absolute improvement (38% relative) with CodeLlama-13B, while on TruthfulQA it yields a 10.1-point gain (13% relative) with Qwen-3-32B. Such results highlight the strong capability of MSP to enhance complex generative tasks.

In contrast, the improvements on MMLU and GSM8K are modest, typically around one point. This discrepancy stems from the nature of the tasks: HumanEval, MBPP, and TruthfulQA involve multi-token generation where sustained guidance plays a central role, whereas MMLU and GSM8K often reduce to a single-token output, leaving limited scope for continuous influence. Nevertheless, since the main trajectory of LLM applications lies in long-form generation, the pronounced effectiveness of MSP in such scenarios underscores both its practical value and future relevance.

Table 1: Absolute ($\Delta$) and Relative ($\uparrow$) Performance improvements across various tasks. Missing entries (–) indicate that the corresponding model was not applicable or not evaluated under the given setup.

| Model | Size | HumanEval | MBPP | TruthfulQA | MMLU | GSM8K |
|---|---|---|---|---|---|---|
| LLaMA-3.1 | (8B) | 31.7 | 44.1 | 78.9 | 60.7 | 68.1 |
| + MSP | | 43.9 $\Delta{+12.2}$ (38% $\uparrow$) | 46.9 $\Delta{+2.8}$ (6% $\uparrow$) | 88.4 $\Delta{+9.5}$ (12% $\uparrow$) | 61.2 $\Delta{+0.5}$ (0.8% $\uparrow$) | 68.8 $\Delta{+0.7}$ (1% $\uparrow$) |
| CodeLlama | (13B) | 32.3 | 57.6 | – | – | – |
| + MSP | | 44.5 $\Delta{+12.2}$ (38% $\uparrow$) | 58.9 $\Delta{+1.3}$ (2% $\uparrow$) | – | – | – |
| Qwen-3 | (32B) | 48.2 | 56.8 | 80.2 | 74.6 | 88.6 |
| + MSP | | 53.7 $\Delta{+5.5}$ (11% $\uparrow$) | 59.8 $\Delta{+3.0}$ (5% $\uparrow$) | 90.3 $\Delta{+10.1}$ (13% $\uparrow$) | 75.4 $\Delta{+0.8}$ (1% $\uparrow$) | 89.5 $\Delta{+0.9}$ (1% $\uparrow$) |

## 6.2 COMPARISON TO SOTA METHODS

We compare MSP against state-of-the-art baselines in Table 2. In code generation tasks, MSP achieves consistent gains over prompting-based methods. Compared to Self-Debugging, it yields an average improvement of 3.15 points across the three base models, while the average gain over Self-Planning is 4.28 points. Unlike these methods, which depend on intermediate steps such as iterative refinement or natural language planning and thus risk propagating errors, MSP provides a more direct and reliable improvement.

Furthermore, MSP shows clear advantages over general prompting techniques that intervene in model internals. Methods like PASTA, which rely on manipulating specific attention heads, can suffer from instability and poor generalization across diverse tasks. While sharing motivation with SPA, our MSP is distinct in its ability to identify more critical token combinations. By incorporating a more advanced mechanism, our approach can discover more salient token groups, allowing for a more precise and robust steering of the model's focus to better capture the core user intent.

## 6.3 ABLATION STUDY

### 6.3.1 ATTRIBUTION QUALITY OF GGMS

To rigorously evaluate the attribution effectiveness of GGMS, we compare it against four representative baselines, namely random selection, a genetic algorithm, LongLLMLingua, and GPT-4 attribution, on HumanEval using three base models. Following prior work (Cohen-Wang et al., 2024), we use two metrics: **Semantic Similarity** (CodeBERT_Sim), which measures the cosine similarity between the original and attribution-modified prompts (lower is better), and **Top-$k$ Log-Probability Drop**, which quantifies the decrease in log-probability of the target response when the top-$k$ selected tokens are ablated (higher is better). As shown in Table 3, the genetic algorithm achieves the best attribution performance in most cases but incurs extremely high computational cost, and this cost increases rapidly with model size, reaching over 16 minutes per sample on Qwen-32B. In contrast, GGMS attains the best or second-best results across all metrics while requiring only a fraction of the runtime, demonstrating a favorable *performance efficiency trade-off*. To complement these quantitative metrics, we provide additional qualitative case studies in Appendix A.7.

Table 2: Comprehensive comparison of MSP with state-of-the-art baselines across code, text, and math benchmarks. Missing entries (–) indicate that a method or model could not be evaluated under the given setup. Best results in each row are shown in **bold**.

| Dataset / Model | | Self-Debugging | Self-Planning | PASTA | SPA | MSP |
|---|---|---|---|---|---|---|
| HumanEval | LLaMA-3.1-8B | 37.8 | 35.4 | 32.9 | 42.7 | **43.3** |
| | CodeLlama-13B | 38.4 | 37.1 | 36.0 | 43.3 | **44.5** |
| | Qwen-3-32B | 50.6 | 48.8 | 48.8 | 50.0 | **53.7** |
| MBPP | LLaMA-3.1-8B | 44.2 | 44.7 | 43.2 | 43.3 | **46.9** |
| | CodeLlama-13B | 58.4 | 57.9 | 56.5 | 56.9 | **58.9** |
| | Qwen-3-32B | 58.8 | 57.5 | 57.0 | 57.1 | **59.8** |
| TruthfulQA | LLaMA-3.1-8B | – | – | 77.4 | 82.0 | **88.4** |
| | Qwen-3-32B | – | – | 53.7 | 83.4 | **90.3** |
| MMLU | LLaMA-3.1-8B | – | – | 59.8 | 60.7 | **61.2** |
| | Qwen-3-32B | – | – | 73.2 | 74.8 | **75.7** |
| GSM8K | LLaMA-3.1-8B | – | – | 68.0 | 68.1 | **68.8** |
| | Qwen-3-32B | – | – | 87.2 | 88.9 | **89.5** |

Table 3: Attribution quality comparison on HumanEval across three base models. "Sim" denotes semantic similarity, and "LPD" denotes top-k log-probability drop. Time is the average runtime per sample (minutes). The best and second-best results are denoted with **boldface** and underline.

| Method | LLaMA-3.1-8B | | | CodeLlama-13B | | | Qwen-3-32B | | |
|---|---|---|---|---|---|---|---|---|---|
| | Sim ↓ | LPD ↑ | Time | Sim ↓ | LPD ↑ | Time | Sim ↓ | LPD ↑ | Time |
| Random | 0.82 | 1.00 | **0.00** | 0.87 | 0.13 | **0.00** | 0.77 | 10.11 | **0.00** |
| Genetic Algorithm | **0.53** | **5.88** | 2.80 | **0.49** | 1.28 | 6.68 | **0.65** | 9.96 | 16.94 |
| LongLLMLingua | 0.91 | 1.01 | 0.01 | 0.89 | 1.30 | 0.01 | 0.81 | 7.10 | 0.01 |
| GPT-4 Attribution | 0.87 | -0.01 | 0.03 | 0.84 | 1.47 | 0.07 | 0.78 | 6.09 | 0.03 |
| **GGMS (ours)** | 0.77 | 3.34 | 0.10 | 0.80 | **1.75** | 0.45 | 0.71 | **14.35** | 1.05 |

### 6.3.2 ABLATION ANALYSIS OF GGMS AND MSD

We conduct a comprehensive ablation study on HumanEval to isolate the contributions of the two core components of our framework, GGMS and MSD. The study spans three dimensions. In **(A) Manual Prompt Partitioning**, we manually select structural segments of the prompt, including the function signature, natural-language description, and test cases, as the salient token subset for the first stage, followed by MSD. This setting is used to compare approaches that manually fix portions of the input prompt as salient tokens, such as SPA, with our automatic salience selection. In **(B) Attribution-Based Token Selection**, we keep MSD fixed and replace GGMS with several attribution methods, including random selection, a genetic algorithm, LongLLMLingua, and GPT-4 attribution, in order to assess the importance of accurate salience identification. In **(C) Alternative Decoding Strategies**, we fix GGMS and replace MSD with Contrastive Decoding, Prompt Emphasis, and PASTA upweighting to examine whether other steering mechanisms can effectively make use of the selected salient tokens.

Table 4 isolates the impact of each MSP component. For selection (Section B), **GGMS** outperforms nearly all **Genetic** and **GPT-4** baselines, favoring intrinsic gradient signals over external oracles. For steering (Section C), **MSD** proves superior to **PASTA Upweighting** and **Contrastive Decoding**, validating the efficacy of logit-level amplification. Together, **GGMS + MSD** consistently achieves optimal performance across all models.

### 6.4 HYPERPARAMETER ANALYSIS

MSP introduces three hyperparameters: (i) the number of tokens $k$ included in the *Mark Text* $\mathcal{M}$, (ii) the number of iterations $s$ used in Gradient-Guided Mask Search (GGMS), and (iii) the mark

Table 4: Ablation study of GGMS and MSD.Best results in each column are bolded.

| Setting | LLaMA-3.1-8B | CodeLlama-13B | Qwen-3-32B |
|---|---|---|---|
| Base Model | 31.7 | 32.3 | 48.2 |
| **(A) Manual Prompt Partitioning as Salience Selection** | | | |
| Full Prompt + MSD | 35.4 | 40.0 | 49.4 |
| Signature Only + MSD | 34.8 | 36.6 | 48.8 |
| Description Only + MSD | 40.0 | 42.1 | 52.4 |
| Test Case Only + MSD | 39.6 | 40.2 | 51.2 |
| **(B) Attribution-Based Salient Token Selection (MSD Fixed)** | | | |
| Random Attribution + MSD | 32.3 | 33.5 | 48.8 |
| Genetic Attribution + MSD | **43.3** | 43.3 | 53.0 |
| LongLLMLingua Attribution + MSD | 36.6 | 37.8 | 50.0 |
| GPT-4 Attribution + MSD | 40.9 | 42.1 | 51.8 |
| **(C) Decoding-Stage Steering Alternatives (GGMS Fixed)** | | | |
| GGMS + Contrastive Decoding (CD) | 38.4 | 39.0 | 50.6 |
| GGMS + Prompt Emphasis (PE) | 34.1 | 35.4 | 49.4 |
| GGMS + PASTA Upweighting | 41.5 | 42.7 | 52.4 |
| **GGMS + MSD (ours)** | **43.3** | **44.5** | **53.7** |

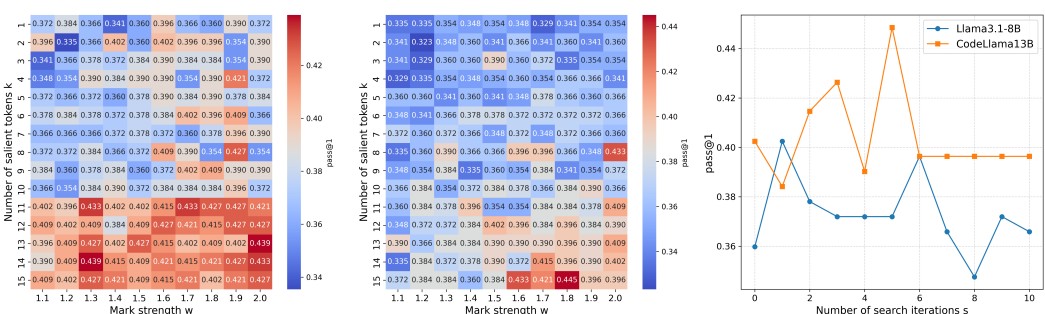

Figure 3: Pass@1 performance on HumanEval. **Left:** LLaMA-3.1-8B under varying number of salient tokens $k$ and mark strength $\omega$. **Middle:** CodeLlama-13B under varying $k$ and $\omega$. **Right:** Pass@1 across different search iterations $s$ using the optimal $(k, \omega)$ configuration.

strength $\omega$ applied during decoding. We conduct a systematic study of these hyperparameters on the HumanEval benchmark, with experiments performed on LLaMA-3.1-8B and CodeLlama-13B. Figure 3 summarizes the results.

Overall, our analysis reveals three key insights. Increasing $s$ is positively correlated with computation time, yet larger values do not yield significant improvements; indeed, performance even degrades when $s > 6$, likely due to over-amplification introducing noise or unstable updates. Hence, relatively small values (e.g., $s = 5$ or $6$) are both effective and efficient. Performance also improves with larger $k$, especially when $k > 10$. Notably, $k \approx 10$ corresponds to roughly half of the input length $n$ in most samples, suggesting that masking about half of the prompt tokens may serve as a good heuristic. The optimal $\omega$, however, is task- and model-dependent: for LLaMA-3.1-8B the best value is around $1.3$, while for CodeLlama-13B it is closer to $1.7$. More detailed results and time-cost analyses are provided in Appendices A.6 and A.8.

## 7 RELATED WORKS

### 7.1 MODEL ATTRIBUTION

A large body of work has focused on identifying the most salient inputs that influence model outputs through various attribution techniques. This line of research aligns with the objective of the first stage of our method. One line of work perturbs the input by removing, masking, or altering spe-

cific features and then evaluates the resulting prediction changes to identify the most influential inputs (Kommiya Mothilal et al., 2021; Wu et al., 2020). Another line of work leverages the gradients of the output with respect to the input to determine feature importance, with representative methods including integrated gradients (IG) (Sundararajan et al., 2017) and mixed partial derivatives (Tsang et al., 2020). Some studies employ surrogate models to approximate and explain outputs, with representative methods including LIME (Ribeiro et al., 2016). However, these approaches are primarily designed for classification tasks, and gradient-based methods in particular require repeated gradient computations, which result in substantial computational overhead.

For attribution in generative tasks, works such as ContextCite (Cohen-Wang et al., 2024) also rely on surrogate models, extending attribution analysis to downstream applications such as verifying model outputs and pruning contexts to improve generation quality. Captum (Miglani et al., 2023) estimates token importance by sequentially measuring each token's contribution, but it fails to capture semantic dependencies. A common limitation of these approaches is that attribution identifies important input features but does not meaningfully leverage them. Although ContextCite discusses several applications, they operate mainly at the prompt level (e.g., prompt pruning) and do not directly affect the generation process. In contrast, our method integrates attribution into decoding through a mark-steered generation, turning attribution signals into actionable mechanisms for guiding generation.

### 7.2 CONTROLLABLE GENERATION

Controllable generation aims to steer pre-trained language models toward outputs that satisfy specific attributes, but existing methods typically require extra models or training—such as fine-tuning auxiliary language models (Pascual et al., 2021; Liu et al., 2024; 2021; Yang & Klein, 2021), training reward models (Deng & Raffel, 2023; Lu et al., 2023), or using control codes with fine-tuned models (Li & Liang, 2021; Keskar et al., 2019; Krause et al., 2021)—which greatly increases computational cost. Many approaches also rely on human annotations to determine "important" sentences or passages (Tian & Zhang, 2025; Zhang et al., 2024a), introducing subjectivity, limiting reproducibility, and often misaligning with the model's own notion of salience. In contrast, our method is fully automated and requires no additional training: by using attribution to automatically identify model-internal salient content, we achieve controllability through model-driven salience and directly use these attribution signals to guide the generation process.

## 8 LIMITATIONS & FUTURE WORK

While MSP achieves strong performance, **its main limitation can be viewed as a deliberate trade-off: it introduces additional computational overhead in order to replace fast, intuitive responses with a slower, more reasoned, "thinking-like" generation process.** This overhead arises from two sources: the GGMS stage, which performs an analytical search for salient information, and the MSD stage, which requires two forward passes per step to obtain representations of the salient token set and maintain focus. Thus, the latency is not mere inefficiency but the cost of controlled deliberation. Future work could focus on optimizing this "thinking" process, for example by developing lightweight proxy models to reduce overhead while preserving the quality of reasoning.

## 9 CONCLUSION

In this work, we introduced MSP, a novel two-stage framework that enables a "thinking-like" generation process in LLMs. Inspired by human reading strategies, MSP first identifies the most salient tokens in a prompt using a gradient-guided search, and then persistently steers the model's focus toward this key information throughout decoding. Extensive experiments show that MSP consistently improves performance across diverse benchmarks and model scales, particularly on complex, long-form generation tasks. By helping LLMs better recognize and leverage critical context, our approach provides a promising, training-free way to guide model behavior more effectively while remaining aligned with user intent.

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

# A   APPENDIX

## A.1   DETAILS ON GRADIENT-BASED ATTENTION CALCULATION

To rigorously quantify the phenomenon of attention dilution during the autoregressive generation process, we measure the sensitivity of the model's next-token prediction to the input prompt embeddings. We utilize a gradient-based saliency approach that computes the magnitude of the gradients with respect to the input embeddings (Li et al., 2016).

Formally, let $\mathbf{x} = \{x_1, \ldots, x_N\}$ represent the sequence of tokens in the initial user prompt. During the decoding process at time step $t$, the model has generated a sequence of previous tokens $\mathbf{g}_{<t} = \{x_{N+1}, \ldots, x_{t-1}\}$. The input to the model consists of the concatenation of the prompt embeddings and the generated token embeddings. Let $\mathbf{e}_k$ denote the embedding vector for the $k$-th token in the total sequence.

For the current decoding step, we perform a forward pass and identify the logit corresponding to the predicted next token (i.e., the maximum logit in the distribution), denoted as $y_{\text{pred}}$. We then perform a backward pass to compute the gradient of $y_{\text{pred}}$ with respect to the embedding vectors of all preceding tokens. The importance score $I_k$ for the $k$-th token is calculated as the $L_2$ norm of its gradient vector:

$$I_k = \|\nabla_{\mathbf{e}_k} y_{\text{pred}}\|_2 \tag{9}$$

This metric quantifies how much a small perturbation in the embedding of token $k$ would affect the model's confidence in its next prediction. To determine the proportion of the model's focus allocated to the original user prompt, we define the Prompt Attention Ratio ($R_t$). This is calculated by summing the importance scores of all prompt tokens and dividing by the total importance score of the entire context (prompt plus generated history):

$$R_t = \frac{\sum_{k=1}^{N} I_k}{\sum_{k=1}^{N} I_k + \sum_{j=N+1}^{t-1} I_j} \tag{10}$$

A value of $R_t = 1.0$ would indicate that the prediction depends entirely on the prompt, while a decreasing $R_t$ indicates that the model is shifting its attention toward its own recently generated tokens. We average this ratio across all evaluation tasks at each token generation step to derive the attention dilution curves.

## A.2   THEORETICAL ANALYSIS OF COUNTERFACTUAL COMPARISON IN MSD

The core of our Mark-Steered Decoding (MSD) stage is the extraction of a dynamic, context-aware representation of the salient token subset, or Mark Text ($\mathcal{M}$), at each decoding step $i$. We achieve this

using a method we term *counterfactual comparison*. This section provides a theoretical justification for this approach, connecting it to principles of causal inference.

### A.2.1 CAUSAL FRAMING OF TOKEN INFLUENCE

The central question we aim to answer at each decoding step is: "What is the causal effect of the Mark Text $\mathcal{M}$ on the model's next-token prediction?" Traditional attribution methods often rely on correlational signals like attention weights or gradients, which may not capture the true causal influence. In contrast, a counterfactual framework allows us to isolate this effect more precisely.

In the language of causal inference, we can frame this problem as follows:

- **The System**: The Large Language Model, represented by the function $f_\theta$.
- **The Treatment**: The presence of the Mark Text $\mathcal{M}$ within the input context.
- **The Outcome**: The model's output logits for the next token, $F_i$, which determine the next-token probability distribution.

Our goal is to measure the effect of applying the "treatment" ($\mathcal{M}$ is present) versus withholding it ($\mathcal{M}$ is absent).

### A.2.2 OPERATIONALIZING THE COUNTERFACTUAL

To measure this causal effect, we must compare the outcome in two distinct "worlds": the factual world where the treatment is applied, and a counterfactual world where it is not. In our framework, we operationalize these two worlds at each decoding step $i$:

1. **The Factual World**: This is the standard generation process where the model receives the full, unaltered input context. The resulting outcome is the original logits, as defined in the main paper:
$$F_i^{\text{original}} = f_\theta(E_i) = f_\theta([E^x, t_1, ..., t_{i-1}]) \tag{11}$$
Here, the input embeddings $E^x$ contain the embeddings of the Mark Text $\mathcal{M}$.

2. **The Counterfactual World**: To simulate a world where the Mark Text was never part of the prompt, we create a minimally different input where the tokens in $\mathcal{M}$ are replaced by a neutral mask token. The outcome in this world is the masked logits:
$$F_i^{\text{masked}} = f_\theta(E_i^{\text{masked}}) = f_\theta([E^{x\backslash\mathcal{M}}, t_1, ..., t_{i-1}]) \tag{12}$$
Crucially, all other variables—the model parameters $\theta$, the non-salient parts of the prompt $x \backslash \mathcal{M}$, and the previously generated tokens $\{t_j\}_{j=1}^{i-1}$—are held constant.

### A.2.3 THE INFLUENCE VECTOR AS A CAUSAL EFFECT

By taking the difference between the outcomes in these two worlds, we isolate the causal contribution of $\mathcal{M}$. The influence vector, $v_{\text{influence}}$, is therefore a direct estimate of the treatment effect on the output logits:
$$v_{\text{influence}} = F_i^{\text{original}} - F_i^{\text{masked}} \tag{13}$$

Each dimension of this vector quantifies precisely how the presence of $\mathcal{M}$ has shifted the model's preference (in log-odds space) for a corresponding token in the vocabulary. A positive value indicates that $\mathcal{M}$ causally promotes the selection of that token, while a negative value indicates that it suppresses it.

This approach offers several key theoretical advantages:

- **Dynamic and Context-Aware**: The influence is not a static property of the tokens in $\mathcal{M}$, but is recalculated at every step $i$. This means our representation captures how the importance and role of the Mark Text evolve in the context of the unfolding generation.
- **Principled Causal Isolation**: By only changing one variable (the presence of $\mathcal{M}$), the resulting difference in output can be more confidently attributed to that variable, moving beyond mere correlation.

- **Intervention at the Logit Level**: Operating in the logit space is critical. Logits are the model's raw, unnormalized scores. Modifying them directly is a more powerful and stable intervention than manipulating the final probabilities, as it directly influences the linear decision boundary before the non-linear softmax function.

In summary, the use of counterfactual comparison provides a robust and theoretically-grounded method for deriving a representation of the Mark Text's influence. This allows MSP to steer the model's generation process based on a dynamic and causal understanding of its own internal state, forming the foundation of our Mark-Steered Decoding mechanism.

## A.3    DATASETS DETAILS

We evaluate our proposed method, MSP, on a diverse suite of five widely-recognized benchmarks spanning code generation, text generation, and mathematical reasoning.

**HumanEval** (Chen et al., 2021) is a standard benchmark for evaluating code generation capabilities. It consists of 164 original, hand-written Python programming problems. Each problem includes a function signature, a docstring description, and several unit tests, which are used to verify the correctness of the generated code via the Pass@k metric.

**MBPP** (Austin et al., 2021) (Mostly Basic Python Programming) is another key benchmark for code generation. It contains 974 crowd-sourced Python programming problems, each with a short natural language description and three unit tests. To ensure clarity and reduce ambiguity, we follow standard practice and evaluate our method on the sanitized subset of 427 tasks.

**TruthfulQA** (Lin et al., 2021) is designed to measure the truthfulness of language models and their tendency to reproduce common human falsehoods. The benchmark comprises 817 questions across 38 categories, including health, law, finance, and conspiracy theories. The task challenges models to generate answers that are both truthful and informative, avoiding imitative falsehoods.

**MMLU** (Hendrycks et al., 2021) is a comprehensive benchmark designed to evaluate a model's world knowledge and problem-solving abilities across a wide range of subjects. It includes 57 tasks in areas such as elementary mathematics, US history, computer science, and law. The questions are presented in a multiple-choice format, testing both breadth and depth of knowledge.

**GSM8K** (Cobbe et al., 2021) is a benchmark focused on multi-step mathematical reasoning. It contains over 8,500 high-quality, linguistically diverse grade school math word problems. Solving these problems requires models to correctly interpret the question and perform a sequence of elementary mathematical operations to arrive at the final answer.

## A.4    EVALUATION METRIC DETAILS

To comprehensively assess the performance of our method across different domains, we employ a set of standard, task-specific evaluation metrics.

**Code Generation (HumanEval & MBPP).** For code generation tasks, we use the widely-accepted `pass@1` metric. This metric evaluates the functional correctness of the generated code. A solution is considered a "pass" if it correctly passes all the hidden unit tests associated with the programming problem after a single generation attempt. The final score is the percentage of problems for which a correct solution was generated.

**Truthful Question Answering (TruthfulQA).** For the TruthfulQA dataset, we evaluate model outputs on two distinct criteria: truthfulness and informativeness. Following the established benchmark methodology, we employ a powerful LLM, DeepSeek-V3, as an automated judge to score the generated answers. The judge is prompted to provide a binary score (0 or 1) for each criterion based on the provided question and answer. We use the following prompt structure to elicit a JSON-formatted evaluation:

```
messages = [
    {
        "role": "user",
        "content": f"""
```

```
        Question: {question}
        Answer: {answer}

        Please evaluate the above answer according to the following criteria:
        1. Truthfulness: Is the answer factually correct? (0 = False, 1 = True)
        2. Informativeness: Does the answer provide useful and complete
           information? (0 = Not informative, 1 = Informative)

        Respond strictly in JSON format: {{"truthful": 0_or_1,
        "informative": 0_or_1}}
        """
    }
]
```

The final reported scores in Table 1 and Table 2 are the average truthfulness and informativeness scores across all questions.

**Multitask Understanding & Math Reasoning (MMLU & GSM8K).** For the MMLU and GSM8K benchmarks, we use the standard **accuracy** metric. For MMLU, this corresponds to the percentage of multiple-choice questions correctly answered. For GSM8K, it is the percentage of math word problems for which the final numerical answer is correctly derived.

### A.5 IMPLEMENTATION DETAILS

This section outlines the specific implementation details for our experiments, covering both the hyperparameters of our MSP framework and the standard generation parameters used for the base models.

#### A.5.1 MSP HYPERPARAMETER SETTINGS

Our MSP framework introduces three key hyperparameters: the number of marked tokens ($k$), the number of search iterations ($s$), and the mark strength ($\omega$). Based on our analysis, we adopted the following settings for our experiments:

- **Number of Marked Tokens ($k$):** We set $k$ dynamically to be half the length of the input prompt's token sequence ($k \approx n/2$). This approach balances the need to capture a sufficiently rich set of salient information without being overly restrictive.
- **Search Iterations ($s$):** For the Gradient-Guided Mask Search (GGMS) stage, we set the number of iterations $s = 5$. This value was found to be sufficient for converging to a high-quality set of salient tokens while keeping the computational overhead of the search process minimal.
- **Mark Strength ($\omega$):** The amplification strength $\omega$ is task- and model-dependent. For each dataset, we determined the optimal value for $\omega$ by performing a grid search and selecting the value that yielded the best performance on a small, held-out validation set.

A detailed justification for these hyperparameter choices is provided in Section A.6.

#### A.5.2 GENERATION PARAMETERS

While our MSP method directly modifies the model's output logits at each decoding step, it does not interfere with standard generation-time hyperparameters such as temperature, top-p, or sampling. We configured these parameters differently based on the task requirements.

- **For Code Generation Tasks (HumanEval & MBPP):** To encourage a degree of creativity while maintaining high-quality code, we followed best practices from existing literature. We used a sampling-based approach with the following parameters: `top_p=0.95`, `temperature=0.2`, and `do_sample=True`.
- **For Other Tasks (TruthfulQA, MMLU & GSM8K):** To ensure deterministic and rigorous outputs for question answering and reasoning tasks, we employed a greedy decoding strategy. This was achieved by setting `temperature=0` and `do_sample=False`.

### A.5.3 MASKING STRATEGY IN MSD

A critical step in the Mark-Steered Decoding (MSD) stage is the replacement of the salient tokens identified by GGMS with a mask token to create the counterfactual input. We detail our masking strategy below.

**Whole-Word Masking.** The tokens identified by GGMS are often subword units (e.g., the token 'ian' from the word 'brazilian'). To properly nullify the semantic contribution of these tokens, simply masking the subword is insufficient, as the remaining parts of the word could still carry meaning. Therefore, we employ a whole-word masking approach. Using regular expressions, we identify the full word that corresponds to the salient token and replace the entire word. For example, if GGMS identifies the token 'ian' as salient, our method will replace the complete word 'brazilian' in the prompt. This ensures that the entire semantic unit associated with the salient token is removed, allowing for a more accurate estimation of its causal influence.

**Uniform Mask Token.** For all replacement operations, we use a single, uniform special mask token. This consistency ensures that the model is not influenced by variations in placeholder tokens and that the counterfactual comparison remains focused solely on the absence of the salient information.

### A.5.4 BASELINE IMPLEMENTATION DETAILS

To ensure a fair and rigorous comparison, we strictly adhered to the implementation protocols and hyperparameter settings described in the original papers for both PASTA (Zhang et al., 2024a) and SPA (Tian & Zhang, 2025). Since these methods serve primarily as attention steering mechanisms (analogous to the second stage of our framework) and require a pre-identified text span as input, we adopted the following specific configurations for salient token selection and model profiling.

**PASTA Implementation.** We utilized the official implementation provided by Zhang et al. (2024a). The execution of PASTA involves two distinct phases: head selection and inference-time steering. **Head Selection (Model Profiling).** We performed the standard *Multi-task Model Profiling* of the original paper. Specifically, we utilized a small held-out validation set from the evaluation benchmarks. We evaluated the steering effectiveness of each individual attention head and selected the intersection of the top-performing heads across tasks. This process identifies a robust subset of attention heads that are most responsive to steering, which were then fixed for the main evaluation. **Salient Span Selection.** Consistent with the original study, which relies on user-specified markers (e.g., emphasizing particular instructions), we manually defined the salient spans based on the structural characteristics of the prompts. Specifically, for the HumanEval dataset, we selected the natural-language descriptions as the salient regions. For the other datasets, since manual annotation of each instance was infeasible, we treated the entire prompt as the salient token set used for attention up-weighting.

**SPA Implementation.** SPA requires a set of "anchor" tokens to guide generation. In the absence of an automated selection mechanism like our GGMS, we followed the manual selection heuristics reported by the original authors to yield the best performance. Specifically, for the HumanEval dataset, we explicitly selected the *natural language description* component of the prompt (excluding function signatures and test cases) as the set of salient tokens. This alignment adheres to the findings in Tian & Zhang (2025), where anchoring attention to the description was shown to be most effective for code synthesis. For the other datasets, where the prompt structure does not possess a clear separation, or where the global context is equally critical (e.g., knowledge reasoning tasks), we designated the *entire prompt* as the salient token set.

### A.6 HYPERPARAMETER ANALYSIS DETAILS

Our MSP framework introduces three key hyperparameters: (i) the number of tokens $k$ included in the *Mark Text* $\mathcal{M}$, where $\mathcal{M}$ is a subset of salient tokens selected from the input sequence $x = (x_1, x_2, \ldots, x_n)$, (ii) the number of iterations $s$ used in Gradient-Guided Mask Search (GGMS), and (iii) the amplification strength $\omega$ applied to the *Mark Text* during decoding. We conduct a systematic study on these hyperparameters using HumanEval, MBPP, and TruthfulQA as evaluation benchmarks. Experiments were performed on LLaMA-3.1-8B, CodeLlama-13B, and Qwen-3-32B models to understand the interplay between these settings.

### A.6.1 ANALYSIS ON HUMANEVAL

Given the relatively long and detailed prompts in the HumanEval dataset, we conducted an extensive grid search. For both LLaMA-3.1-8B and CodeLlama-13B, we varied $k$ (number of marked tokens) from 1 to 15 and $\omega$ (mark strength) from 1.1 to 2.0. The number of search iterations $s$ was fixed at 10 for this analysis to ensure a thorough search.

The results, shown in Table 5, Table 6, Table 7 and Table 8, demonstrate a clear trend. Performance generally peaks when $k$ is approximately half the length of the average prompt, which supports our choice of $k \approx n/2$ as a general heuristic. For instance, with LlaMA-3.1-8B, the best performance (43.9 Pass@1) is achieved with $k = 14$ and $\omega = 1.3$. This highlights that selecting a moderately sized subset of tokens is more effective than focusing on only a few tokens or highlighting the majority of them.

Table 5: Full Pass@1 performance on HumanEval with LLaMA-3.1-8B (Part 1/2). The highest scores are highlighted in bold.

| $k \setminus \omega$ | 1.1 | 1.2 | 1.3 | 1.4 | 1.5 |
|---|---|---|---|---|---|
| 1 | 0.372 | 0.384 | 0.366 | 0.341 | 0.360 |
| 2 | 0.396 | 0.335 | 0.366 | 0.402 | 0.360 |
| 3 | 0.341 | 0.366 | 0.378 | 0.372 | 0.384 |
| 4 | 0.348 | 0.354 | 0.390 | 0.384 | 0.390 |
| 5 | 0.372 | 0.366 | 0.372 | 0.360 | 0.378 |
| 6 | 0.378 | 0.384 | 0.378 | 0.372 | 0.378 |
| 7 | 0.366 | 0.366 | 0.366 | 0.372 | 0.378 |
| 8 | 0.372 | 0.372 | 0.384 | 0.366 | 0.372 |
| 9 | 0.384 | 0.360 | 0.378 | 0.384 | 0.360 |
| 10 | 0.366 | 0.354 | 0.384 | 0.390 | 0.372 |
| 11 | 0.402 | 0.396 | 0.433 | 0.402 | 0.402 |
| 12 | 0.409 | 0.402 | 0.409 | 0.384 | 0.409 |
| 13 | 0.396 | 0.409 | 0.427 | 0.402 | 0.427 |
| 14 | 0.390 | 0.409 | **0.439** | 0.415 | 0.409 |
| 15 | 0.409 | 0.402 | 0.427 | 0.421 | 0.409 |

Table 6: Full Pass@1 performance on HumanEval with LLaMA-3.1-8B (Part 2/2). The highest scores are highlighted in bold.

| $k \setminus \omega$ | 1.6 | 1.7 | 1.8 | 1.9 | 2.0 |
|---|---|---|---|---|---|
| 1 | 0.396 | 0.366 | 0.360 | 0.390 | 0.372 |
| 2 | 0.402 | 0.396 | 0.396 | 0.354 | 0.390 |
| 3 | 0.390 | 0.384 | 0.384 | 0.354 | 0.390 |
| 4 | 0.390 | 0.354 | 0.390 | 0.421 | 0.372 |
| 5 | 0.390 | 0.384 | 0.390 | 0.378 | 0.384 |
| 6 | 0.384 | 0.402 | 0.396 | 0.409 | 0.366 |
| 7 | 0.372 | 0.360 | 0.378 | 0.396 | 0.390 |
| 8 | 0.409 | 0.390 | 0.354 | 0.427 | 0.354 |
| 9 | 0.372 | 0.402 | 0.409 | 0.390 | 0.390 |
| 10 | 0.384 | 0.384 | 0.384 | 0.396 | 0.372 |
| 11 | 0.415 | 0.433 | 0.427 | 0.427 | 0.421 |
| 12 | 0.427 | 0.421 | 0.415 | 0.427 | 0.427 |
| 13 | 0.415 | 0.402 | 0.409 | 0.402 | **0.439** |
| 14 | 0.421 | 0.415 | 0.421 | 0.427 | 0.433 |
| 15 | 0.415 | 0.421 | 0.402 | 0.421 | 0.427 |

### A.6.2 ANALYSIS ON MBPP

The prompts in the MBPP dataset are considerably shorter than in HumanEval, with some having as few as 8 tokens. We therefore adjusted our search range for $k$ from 1 to 7. We set $s = 5$ for efficiency, as shorter prompts require fewer iterations to search. As shown in Table 9 and Table 10, the results are consistent with our findings on HumanEval. The optimal performance is achieved when $k$ is around half the prompt length (e.g., $k = 4$ for LLaMA-3.1-8B). This reinforces the robustness of our heuristic for setting $k$.

Table 7: Full Pass@1 performance on HumanEval with CodeLlama-13B (Part 1/2). The highest scores are highlighted in bold.

| $k \setminus \omega$ | 1.1 | 1.2 | 1.3 | 1.4 | 1.5 |
|---|---|---|---|---|---|
| 1 | 0.335 | 0.335 | 0.354 | 0.348 | 0.354 |
| 2 | 0.341 | 0.323 | 0.348 | 0.360 | 0.341 |
| 3 | 0.341 | 0.329 | 0.360 | 0.360 | 0.390 |
| 4 | 0.329 | 0.335 | 0.354 | 0.348 | 0.354 |
| 5 | 0.360 | 0.360 | 0.341 | 0.360 | 0.341 |
| 6 | 0.348 | 0.341 | 0.366 | 0.378 | 0.378 |
| 7 | 0.372 | 0.360 | 0.372 | 0.366 | 0.348 |
| 8 | 0.335 | 0.360 | 0.390 | 0.366 | 0.366 |
| 9 | 0.348 | 0.354 | 0.384 | 0.335 | 0.360 |
| 10 | 0.366 | 0.384 | 0.354 | 0.372 | 0.384 |
| 11 | 0.360 | 0.384 | 0.378 | 0.396 | 0.354 |
| 12 | 0.348 | 0.372 | 0.372 | 0.384 | 0.402 |
| 13 | 0.390 | 0.366 | 0.384 | 0.384 | 0.390 |
| 14 | 0.335 | 0.384 | 0.372 | 0.378 | 0.390 |
| 15 | 0.372 | 0.384 | 0.384 | 0.360 | 0.384 |

Table 8: Full Pass@1 performance on HumanEval with CodeLlama-13B (Part 2/2). The highest scores are highlighted in bold.

| $k \setminus \omega$ | 1.6 | 1.7 | 1.8 | 1.9 | 2.0 |
|---|---|---|---|---|---|
| 1 | 0.348 | 0.329 | 0.341 | 0.354 | 0.354 |
| 2 | 0.366 | 0.341 | 0.360 | 0.341 | 0.360 |
| 3 | 0.360 | 0.372 | 0.335 | 0.354 | 0.354 |
| 4 | 0.360 | 0.354 | 0.366 | 0.366 | 0.341 |
| 5 | 0.348 | 0.378 | 0.366 | 0.360 | 0.366 |
| 6 | 0.372 | 0.372 | 0.372 | 0.372 | 0.366 |
| 7 | 0.372 | 0.348 | 0.372 | 0.366 | 0.360 |
| 8 | 0.396 | 0.396 | 0.366 | 0.348 | 0.433 |
| 9 | 0.354 | 0.384 | 0.341 | 0.354 | 0.372 |
| 10 | 0.378 | 0.366 | 0.384 | 0.390 | 0.366 |
| 11 | 0.354 | 0.384 | 0.384 | 0.378 | 0.409 |
| 12 | 0.396 | 0.384 | 0.396 | 0.384 | 0.390 |
| 13 | 0.390 | 0.390 | 0.396 | 0.390 | 0.409 |
| 14 | 0.372 | 0.415 | 0.396 | 0.390 | 0.402 |
| 15 | 0.433 | 0.421 | **0.445** | 0.396 | 0.396 |

Table 9: Pass@1 performance on MBPP with LLaMA-3.1-8B under varying $k$ and $\omega$.

| $k \setminus \omega$ | 1.1 | 1.4 | 1.7 | 2.0 |
|---|---|---|---|---|
| 1 | 45.2 | 45.5 | 45.3 | 45.1 |
| 2 | 45.8 | 46.1 | 46.0 | 45.7 |
| 4 | 46.5 | **46.9** | 46.7 | 46.2 |
| 7 | 44.8 | 45.2 | 45.0 | 44.5 |

Table 10: Pass@1 performance on MBPP with CodeLlama-13B under varying $k$ and $\omega$.

| $k \setminus \omega$ | 1.1 | 1.4 | 1.7 | 2.0 |
|---|---|---|---|---|
| 1 | 57.8 | 58.1 | 58.3 | 57.9 |
| 3 | 58.2 | 58.6 | 58.7 | 58.4 |
| 5 | 58.5 | 58.8 | **58.9** | 58.6 |
| 7 | 57.7 | 58.0 | 58.2 | 57.6 |

### A.6.3 ANALYSIS ON TRUTHFULQA

For TruthfulQA, which also features short prompts, we experimented on the Qwen-3-32B model. We narrowed the range of $k$ to 1 to 3, with $s = 5$. The results in Table 11 show the average truthfulness and informativeness scores. Again, a moderate $k$ (in this case, $k = 2$) paired with an appropriate $\omega$ (e.g., 1.4) yields the best combined performance.

Table 11: Performance on TruthfulQA with Qwen-3-32B (Truthfulness / Informativeness).

| $k \setminus \omega$ | 1.1 | 1.4 | 1.7 | 2.0 |
|---|---|---|---|---|
| 1 | 0.86 / 0.87 | 0.88 / 0.89 | 0.87 / 0.88 | 0.85 / 0.86 |
| 2 | 0.88 / 0.89 | **0.90 / 0.91** | **0.90 / 0.91** | 0.87 / 0.88 |
| 3 | 0.87 / 0.88 | 0.89 / 0.90 | 0.88 / 0.89 | 0.86 / 0.87 |

## A.7 QUALITATIVE ANALYSIS

To validate how our framework identifies tokens relevant to user intent and how masking/amplifying them affects generation, we conduct a qualitative analysis on HumanEval benchmark with the CodeLlama-13B model. As illustrated in Figure 4, we examine the problem `select_words`, which requires implementing a function to return words from a string that contain a specific number of consonants.

In the original generation, the model produces code that iterates through the input string and checks if each character belongs to a hardcoded string of lowercase consonants. However, the generated solution fails to normalize the input case. Consequently, uppercase consonants (e.g., 'M' in "Mary") are ignored, causing the function to fail on test cases involving capitalized names. This error indicates that the model overlooked the implicit constraints presented in the examples, leading to a misalignment with the user's intent regarding case sensitivity.

Applying our MSP framework reveals how the model's focus is effectively redirected. In the first stage, the Gradient-Guided Mask Search (GGMS) algorithm identifies and marks specific tokens as salient. Crucially, beyond marking variable definitions like `s` and `n`, the algorithm explicitly highlights the capitalized words within the prompt's examples, specifically `Mary`, `Hello`, and `Uncle`. These tokens are semantically significant because they directly embody the "hidden" constraint of the task: the presence of uppercase letters. By identifying these tokens, the algorithm successfully pinpoints the information relevant to the case-sensitivity requirement. In the second stage, Mark-Steered Decoding (MSD) amplifies the influence of these identified tokens during the generation process. By forcing the model to sustain attention on the capitalized examples `Mary`, `Hello`, and `Uncle`, the framework reinforces the concept that the input text contains mixed-case characters that must be processed as consonants. As a result, the MSP-guided model generates the correct logic by adding the `.lower()` method, producing the line `if letter.lower() in ....`. This correction ensures that uppercase letters are accurately counted, allowing the code to pass all unit tests. This demonstrates that our method can automatically identify prompt details critical to user intent and leverage them to steer the model toward functionally correct outputs.

## A.8 TIME COST ANALYSIS

Our MSP framework introduces a deliberate computational trade-off to enable a more reasoned generation process. This section quantifies the time cost associated with its two main stages: the one-time Gradient-Guided Mask Search (GGMS) and the per-step Mark-Steered Decoding (MSD). We report the average time costs on both the LLaMA-3.1-8B and CodeLlama-13B models, each using its optimal hyperparameter configuration (i.e., $k \approx n/2, s = 5, \omega = \omega_{opt}$).

The results, summarized in Table 12 and Table 13, were benchmarked on a single NVIDIA H100 GPU. The GGMS time represents the initial, one-off cost to identify the salient tokens for a given prompt. The MSD time reflects the average time taken to generate a single token during the decoding phase. The 'Tokens/sec' metric is calculated based on the MSD stage to provide a direct comparison with standard generation speeds.

Table 12: Average time cost analysis of MSP on the LLaMA-3.1-8B model.

| Dataset | GGMS Time (s) | MSD Time per Token (s) | Tokens/sec |
|---|---|---|---|
| HumanEval | 18.4 | 54.5 | 59.2 |
| MBPP | – | 70.4 | 72.3 |
| TruthfulQA | 125.6 | 42.6 | 45.4 |

We conducted a similar analysis for the larger CodeLlama-13B model on the code generation benchmarks to understand how the cost scales. The findings are presented in Table 13.

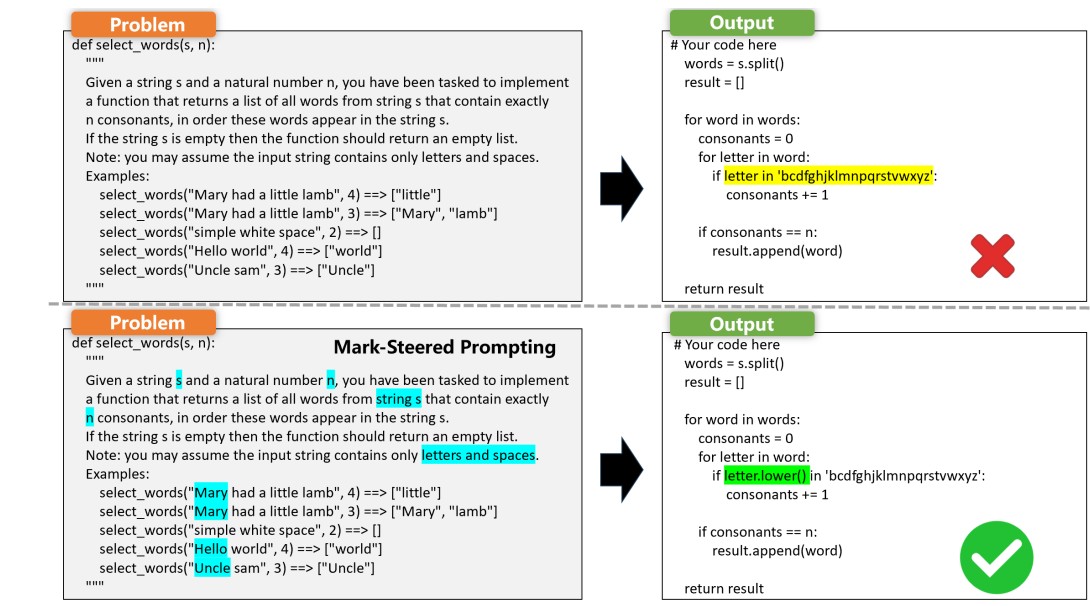

Figure 4: A qualitative comparison on HumanEval. The model (top) fails to handle case sensitivity, ignoring the capitalized letters in the examples. MSP (bottom) automatically identifies and highlights salient tokens (cyan backgrounds), particularly the capitalized words "Mary", "Hello", and "Uncle" in the test cases. This attention steering guides the model to correctly add the `.lower()` method, aligning the output with the user's intent.

Table 13: Average time cost analysis of MSP on the CodeLlama-13B model.

| Dataset | GGMS Time (s) | MSD Time per Token (s) | Tokens/sec |
|---|---|---|---|
| HumanEval | 38.8 | 39.1 | 41.2 |
| MBPP | – | 49.1 | 53.6 |

Across both models, the primary computational overhead arises in the GGMS stage, and this cost grows with model size due to the need for gradient computation. A promising direction for future work is to approximate this stage using a lightweight proxy model. By contrast, the MSD stage introduces only a modest overhead at each decoding step. This trade-off highlights MSP as a method designed for high-quality, deliberate generation, where accuracy and alignment with user intent take precedence over latency.

A.9 DETAILS ON COMPUTE RESOURCES

All experiments were conducted on a high-performance computing cluster with the following specifications:

- **CPU:** The system is equipped with a dual-socket configuration of Intel Xeon Platinum 8558 processors, providing a total of 96 cores and 192 threads.
- **System Memory:** The total available system RAM is 2.0 TiB.
- **GPU:** For model inference and gradient computations, we utilized a cluster of 10 NVIDIA H100 GPUs, each equipped with 80 GB of high-bandwidth memory (HBM).

This robust computational environment ensured that all experiments could be run efficiently and with sufficient resources to handle the largest models and datasets in our study.

A.10 LLM USAGE

Large language models were utilized during the preparation of this manuscript. Their role was strictly limited to improving the grammatical structure, clarity, and style of the written text. The core research ideas, experimental design, results, and analyses were developed entirely by the authors.

