# OpenReview forum: "Salience Aware Mark-Steered Prompting For LLMs"
_ICLR.cc/2026/Conference — ICLR 2026 Conference Withdrawn Submission_

### Official Review · Reviewer_Hfpc · 2025-10-21

**Soundness:** 3
**Presentation:** 3
**Contribution:** 2
**Rating:** 4
**Confidence:** 4

**Summary:**

The authors introduce a 2-stage method to improve model steering. The method first automatically identifies the most influential tokens and then linearly amplifies their influence, using some computed overhead. Results for three LLMs on 4 datasets show some improvements, compared to prompting-based baselines as well as baselines requiring human prompt annotation.

**Strengths:**

- The studied problem is important and timely
- The introduced method seems intuitive
- The paper is well-written and clear throughout
- The performance improvements in Table 1 are substantial

**Weaknesses:**

There seems to be a lack of clarity in the evaluation/baselines. The authors propose a two-step procedure, with both steps being novel (to the best of my knowledge). The first step selects a span of the prompt to emphasize while the second step emphasizes it.

The two non-prompting baselines (PASTA and SPA) they compare against require selecting a span of the prompt to emphasize, i.e. they are designed only for the second step -- how did the authors do this selection (i.e. the first step)? The authors do provide some comparisons of different naive methods for the first step in Table 3 - it would be helpful to see these two methods in this table as well to see whether the second step is actually providing gains or whether the gains come from just omitting the first step. Additionally, it might be nice to see a less restrictive way of selecting the span, i.e. prompting the model to to select the span before emphasizing it (as is done in AutoPASTA https://arxiv.org/abs/2409.10790). It would additionally be nice to understand more details about how the baselines were run (e.g. how was head selection conducted for PASTA?)

While the authors do show some ablations for Step 1, it would be nice also to see ablations comparing against different methods for emphasizing the selected span in Step 2. Examples include contrastive decoding, prompt emphasis (e.g. adding asterisks or context-faithful prompting: https://arxiv.org/abs/2303.11315), or pasta upweighting.

**Questions:**

See above

---

> ### Author Response · Authors · 2025-11-22
> **Response to Reviewer Hfpc(1/2)**
>
> We thank the reviewer for the insightful comment.
>
> **1. Clarification on PASTA and SPA Implementation**
>  The reviewer is correct that PASTA and SPA are primarily steering mechanisms (Step 2) that require a pre-identified text span. In our main comparison (Table 2), **we strictly follow the original papers when performing the first-stage selection of salient tokens.**
>
> - **Selection Strategy for SPA:** For SPA, the first-stage selection is entirely based on manual identification. Therefore, for HumanEval, we select the natural-language description part of the prompt as the set of salient tokens—this is also the manually chosen setting reported to yield the best performance in the SPA paper. For other datasets, since there is no clear separation within the prompt, we use the entire prompt as the salient token set.
> - **Head Selection for PASTA:** Following the PASTA paper, we performed the standard **model profiling** phase using a small validation set to identify the most effective attention heads for steering.
> - **Revisions:** We have added a detailed "Baseline Implementation Details" section in the **Appendix.A.5.4 of the revised manuscript**, fully documenting the span selection heuristics and head profiling procedures.

---

> ### Author Response · Authors · 2025-11-22
> **Response to Reviewer Hfpc(2/2)**
>
> **2. Comprehensive Component-Wise Ablations (Revised Tables 4B & 4C)**
>
> To rigorously isolate the gains of our two stages—**Gradient-Guided Mask Search (GGMS)** and **Mark-Steered Decoding (MSD)**—and to address your suggestion of comparing against "AutoPASTA" selection and "Prompt Emphasis" steering, we have conducted extensive new ablations.
>
> **(A) Evaluation of Step 1: Attribution-Based Salient Token Selection (MSD Fixed)**
>
> To validate GGMS, we fixed the decoding strategy (MSD) and replaced GGMS with four alternative selection methods. The results are reported in **Revised Table 4 (B)**:
>
> 1. **Random Selection:** Randomly selecting tokens (as a lower bound).
> 2. **Genetic Algorithm (GA):** A heuristic search similar to EvoPrompt [1].
> 3. **LongLLMLingua[2]:** A state-of-the-art prompt compression method used to select key tokens.
> 4. **GPT-4 Attribution:** We prompted GPT-4 to identify the most salient keywords or constraints in the prompt. This setting effectively serves as a proxy for the “AutoPASTA” approach, as requested by the reviewer. We note that we adopted this proxy because the official AutoPASTA code repository is no longer accessible (the provided link returns a 404 error).
>
> | **Setting**                     | **LLaMA-3.1-8B** | **CodeLlama-13B** | **Qwen-3-32B** |
> | ------------------------------- | ---------------- | ----------------- | -------------- |
> | Random Attribution + MSD        | 32.3             | 33.5              | 48.8           |
> | Genetic Attribution + MSD       | **43.3**         | 43.3              | 53.0           |
> | LongLLMLingua Attribution + MSD | 36.6             | 37.8              | 50.0           |
> | GPT-4 Attribution + MSD         | 40.9             | 42.1              | 51.8           |
> | **GGMS + MSD (ours)**           | **43.3**         | **44.5**          | **53.7**       |
>
> Experimental results demonstrate that GGMS consistently outperforms nearly all baseline methods. While the Genetic Algorithm (GA) occasionally achieves performance parity with GGMS, it incurs a prohibitive computational overhead(as shown in Table 3 of our revised manuscript), rendering it impractical for real-world applications. Furthermore, GGMS proves superior to both LLM-based attribution (e.g., GPT-4 Attribution) and prompt compression techniques (e.g., LongLLMLingua). The advantage over LLM attribution stems from the fact that such methods rely solely on the model’s explicit assessment without accessing gradients or internal signals, rendering them inherently unreliable. Similarly, prompt compression algorithms typically utilize a smaller proxy model (specifically Llama-2-7b-chat, following the standard LongLLMLingua setting) rather than the target model itself, resulting in a misalignment between the proxy’s selection and the target model’s actual needs.
>
> **(B) Evaluation of Step 2: Decoding-Stage Steering Alternatives (GGMS Fixed)**
>
> To validate MSD, we fixed the selection method (GGMS) and applied different steering strategies to the identified tokens. The results are reported in **Revised Table 4 (C)**:
>
> 1. **Prompt Emphasis:** We wrapped the GGMS-identified tokens in asterisks (e.g., `*token*`) to simulate "context-faithful prompting."
> 2. **Contrastive Decoding (CD):** We applied standard Contrastive Decoding penalties to the identified spans.
> 3. **PASTA Upweighting:** We applied the PASTA attention-steering mechanism  specifically to the tokens identified by GGMS.
>
> | **Setting**                      | **LLaMA-3.1-8B** | **CodeLlama-13B** | **Qwen-3-32B** |
> | -------------------------------- | ---------------- | ----------------- | -------------- |
> | GGMS + Contrastive Decoding (CD) | 38.4             | 39.0              | 50.6           |
> | GGMS + Prompt Emphasis (PE)      | 34.1             | 35.4              | 49.4           |
> | GGMS + PASTA Upweighting         | 41.5             | 42.7              | 52.4           |
> | **GGMS + MSD (ours)**            | **43.3**         | **44.5**          | **53.7**       |
>
> Experimental results show that our proposed **GGMS + MSD** combination achieved the highest performance. While applying PASTA to GGMS tokens (PASTA Upweighting) improved over standard prompting, MSD provided superior steering. This suggests that intervening at the **logit level** (MSD) is more effective for fine-grained token steering than intervening at the **attention head level** (PASTA), which may suffer from head-selection noise.
>
> **Conclusion** These new experiments confirm that the gains are not derived solely from one step. Both the precise identification of salient tokens (GGMS) and the robust logits-based steering (MSD) are essential. We believe these added comparisons provide the clarity and depth requested.
>
> [1]Connecting Large Language Models with Evolutionary Algorithms Yields Powerful Prompt Optimizers(ICLR'24)
>
> [2] LongLLMLingua: Accelerating and Enhancing LLMs in Long Context Scenarios via Prompt Compression (ACL' 24)

---

> > ### Comment · Reviewer_Hfpc · 2025-11-25
> >
> > I thank the reviewers for their response and particularly the new experiments. I have updated my score from 4 to 6 accordingly.

---

> > > ### Author Response · Authors · 2025-11-28
> > > **Appreciation for Reviewer Hfpc’s Feedback**
> > >
> > > Thank you very much, Reviewer Hfpc, for your thoughtful feedback and for updating your score. We truly appreciate your recognition of our additional analysis and experiments, and we are grateful for your constructive insights throughout the review process.

---

### Official Review · Reviewer_XiCK · 2025-10-22

**Soundness:** 2
**Presentation:** 3
**Contribution:** 2
**Rating:** 2
**Confidence:** 4

**Summary:**

This paper investigates the problem of identifying salient tokens in language model inputs, which are tokens that are most critical for generation quality and alignment with user intent. The authors propose an algorithm that uses gradient-based updates to find which input tokens, when masked, most significantly impact the model's output probability distribution. They then amplify these identified salient tokens by scaling their logits during generation.

Although interesting, the paper has several significant weaknesses. The gradient-based masking approach treats the fundamentally discrete mask as continuous, making it unclear whether high gradients reliably indicate importance. The connection to user intent is also poorly validated since the paper lacks qualitative analysis showing which tokens are actually identified as salient and how masking them affects generated outputs. Most importantly, the method is computationally expensive, requiring multiple forward and backward passes with three task-dependent hyperparameters (s, k, w) whose optimal values must be tuned per model and task. Hence, this raises questions about whether simpler alternatives like chain-of-thought prompting might achieve similar results more efficiently.

**Strengths:**

- The problem of salient token identification is an interesting and timely topic.
- The paper is well written and easy to follow.

**Weaknesses:**

- **Algorithm:** The proposed algorithm identifies salient tokens by computing gradients with respect to a mask. However, the mask is treated as continuous, even though it is fundamentally combinatorial (as seen in Eq. (2)). It is unclear how high or low gradients reliably indicate token importance, as a gradient could be high at m = 1 but result in a better loss at m = 0 in Eq. (3). Additionally, the authors refine the mask by substituting masked and unmasked tokens one at a time and evaluating Eq. (3). This choice is not justified, particularly compared to other well-established binary optimization frameworks, such as genetic algorithms.

- **Relationship with user intent:** The framework claims to identify tokens most relevant to user intent (page 5). However, it is difficult to see how the algorithm achieves this. A qualitative analysis is necessary: which tokens are identified as salient by the algorithm, and how does masking them affect the generated outputs? The current focus on quantitative results alone does not fully validate the framework. Moreover, masking tokens just based on the probability of generation of a sequence could have unintended consequences, such as slightly changing generated tokens while significantly impacting meaning or factual correctness. This raises concerns about the objective function in Eq. (3), which requires further justification (especially qualitative one).

- **Complexity:** The method relies on masking input tokens and performing multiple forward and backward passes across many mask combinations, particularly during s gradient-guided mask updates. After identifying k (a hyperparameter) salient tokens, another hyperparameter w scales their logits. The optimal values of s, k, and w are task- and model-dependent (page 9), which introduces a high computational overhead. It is unclear whether simpler approaches, such as chain-of-thought prompting, could achieve comparable results with similar or even lower computational cost.

- **Rigor:** In Eq. (3), m is defined as {0,1}^T, but since it is applied via a Hadamard product with x, I imagine it should be {0,1}^n. Additionally, the claim that “during autoregressive generation, the model tends to assign equal attention to all input tokens” is inaccurate. Transformers compute attention scores based on token embeddings, so the model inherently encodes information about token importance. Therefore, the claim in Section 3 is misleading.

- **Minor typo:** In Eq. (2), x should be bolded as it represents a vector of input tokens.

**Questions:**

Please refer to Weaknesses section.

---

> ### Author Response · Authors · 2025-11-22
> **Response to Reviewer XiCK(1/4)**
>
> ## Weakness 1
>
> We thank the reviewer for the insightful comments.
>
> We acknowledge the reviewer's insight regarding the combinatorial nature of token masking and the potential discrepancy between local gradients (at $m=1$) and global loss changes (at $m=0$). However, we respectfully argue that gradients remain a robust proxy for token saliency for the following reasons:
>
> 1. **Gradients Reflect Sensitivity:** As noted in the comprehensive survey by Zhao et al. (2024) [1], gradient-based attribution methods operate on the principle that the magnitude of partial derivatives reflects the **sensitivity** of the model's output to specific input features. A high gradient at $m=1$ indicates that the model is highly sensitive to the presence of that token; effectively, the local slope serves as a directional indicator for the "steepest descent" in information loss.
> 2. **Continuous Relaxation is Standard Practice:** While tokens are discrete, treating the mask as continuous during the search phase allows us to leverage efficient differentiable optimization to navigate the vast combinatorial search space. The gradient provides the necessary heuristic to identify the subset of tokens that—if removed—would most likely degrade performance, serving as a high-quality initialization for subsequent refinement steps.
>
> Regarding the justification of our refinement strategy and the comparison with binary optimization frameworks, we conducted a comprehensive quantitative evaluation. To evaluate the quality of masks produced by GGMS, we followed the protocol of Cohen-Wang et al. (2024) [2] and adopted two metrics:
>
> (1) Semantic Similarity (CodeBERT Sim), measuring the semantic distance between original and masked prompts (lower is better), and(2) Top-k Log-Probability Drop (LDP), measuring how much the target output probability decreases when selected tokens are ablated (higher is better).
>
> We compared GGMS against four baselines: Random,Genetic Algorithm (GA), LongLLMLingua[3], and GPT-4 Attribution.The results  reveal a critical trade-off. While the Genetic Algorithm (GA) yields strong attribution quality, it incurs prohibitive computational costs—requiring over **16 minutes per sample** on Qwen-32B, rendering it impractical for real-world applications. Conversely, LongLLMLingua and GPT-4 Attribution are efficient but perform poorly on attribution metrics; notably, GPT-4 Attribution exhibited instability, producing reversed attributions (negative Drop scores) in some settings.
>
> In comparison, **GGMS achieves the best trade-off between attribution quality and computational efficiency**. It significantly outperforms efficient baselines in LDP while remaining orders of magnitude faster than GA. We have added these full results and discussion to **Section 6.3.1** of the revised manuscript.
>
> |                   |          | **LLaMA-3.1-8B** |          |          | **CodeLlama-13B** |          |          | **Qwen-3-32B** |          |
> | ----------------- | :------: | :--------------: | :------: | :------: | :---------------: | :------: | :------: | :------------: | :------: |
> | **Method**        |  Sim ↓   |       LDP↑       |   Time   |  Sim ↓   |       LDP↑        |   Time   |  Sim ↓   |      LDP↑      |   Time   |
> | Random            |   0.82   |       1.00       | **0.00** |   0.87   |       0.13        | **0.00** |   0.77   |     10.11      | **0.00** |
> | Genetic Algorithm | **0.53** |     **5.88**     |   2.80   | **0.49** |       1.28        |   6.68   | **0.65** |      9.96      |  16.94   |
> | LongLLMLingua     |   0.91   |       1.01       |  *0.01*  |   0.89   |       1.30        |  *0.01*  |   0.81   |      7.10      |  *0.01*  |
> | GPT-4 Attribution |   0.87   |      -0.01       |   0.03   |   0.84   |      *1.47*       |   0.07   |   0.78   |      6.09      |   0.03   |
> | **GGMS (ours)**   |  *0.77*  |      *3.34*      |   0.10   |  *0.80*  |     **1.75**      |   0.45   |  *0.71*  |   **14.35**    |   1.05   |
>
> [1] Explainability for Large Language Models: A Survey (ACM Transactions on Intelligent Systems and Technology, 2024).
>
> [2] ContextCite: Attributing Model Generation to Context.(NeurIPS'24)
>
> [3]LongLLMLingua: Accelerating and Enhancing LLMs in Long Context Scenarios via Prompt Compression (ACL'24)

---

> ### Author Response · Authors · 2025-11-22
> **Response to Reviewer XiCK(2/4)**
>
> ## Weakness 2
>
> We thank the reviewer for this insightful comment.
>
> 1、Inclusion of Qualitative Analysis
>
> We have added a detailed Qualitative Analysis section in the Appendix.A.7 of our revised manuscript. In this new section, we provide concrete case studies displaying:
>
> - **Identified Salient Tokens:** Visualizations of the tokens selected by the Gradient-Guided Mask Search (GGMS). These examples demonstrate that the algorithm consistently identifies constraints and key instructional verbs that are central to the user's intent, rather than random high-probability tokens.
> - **Effect on Output:** Side-by-side comparisons showing how the generated outputs differ before and after applying MSP. These qualitative examples highlight how masking-based steering effectively corrects specific failure modes by reinforcing the model's focus on the identified salient spans.
>
> 2、Clarification on the "Unintended Consequences" of Masking
>
> Regarding the concern that masking might lead to "unintended consequences" or affect factual correctness during generation, we respectfully wish to clarify a misunderstanding regarding how the mask is utilized in our framework.
>
> **We emphasize that the model never generates tokens based on the masked input.** The masking mechanism is employed strictly to *estimate* token influence, not to construct the context used for the actual generation.
>
> Specifically, during the decoding step $i$:
>
> 1. Base Inference: The model computes its actual output distribution using the **full, unmasked input** $E_i$. This produces the original logits, $F_i^{\text{original}}$.
>
> 2. Influence Estimation: We perform a parallel forward pass using the masked version $E_i^{\text{masked}}$ solely to compute the difference in logits:
>
>    $$v_{\text{influence}} = F_i^{\text{original}} - F_i^{\text{masked}}$$
>
>    This vector $v_{\text{influence}}$ captures the causal contribution of the salient token set.
>
> 3. Final Generation: As shown in Eqs. (5)–(8) of the paper, the final logits used for decoding are:
>
>    $$F_i^{\text{augmented}} = F_i^{\text{original}} + \omega \cdot v_{\text{influence}}$$
>
> Because the base term is $F_i^{\text{original}}$, the model’s generation is always grounded in the complete, grammatically correct, and factually intact context $E_i$. The $v_{\text{influence}}$ term merely acts as a steering signal to amplify attention to specific instructions. Therefore, no masked or corrupted input is ever provided to the model for the purpose of next-token selection, ensuring that the semantic integrity and factual correctness of the prompt are preserved.

---

> ### Author Response · Authors · 2025-11-22
> **Response to Reviewer XiCK(3/4)**
>
> ## Weakness 3
>
> We would like to clarify the practical tuning burden of our hyperparameters and demonstrate, through new experiments, that MSP is orthogonal to CoT and serves as a powerful enhancement rather than a competitor.
>
> 1. Clarification on Hyperparameter Complexity
>
> While our method involves three hyperparameters ($s, k, \omega$), our empirical analysis indicates that the actual tuning burden is much lighter than it appears. The "search space" is effectively constrained by robust heuristics:
>
> - **$s$ (Search Iterations):** As illustrated in **Figure 3 (Right)** of our revised manuscript, the model's performance is not sensitive to $s$ once it reaches a small threshold. Our results show that a small value (e.g., $s=5$) is sufficient for convergence, and larger values do not yield significant gains. Therefore, $s$ can be treated as a **fixed, low-cost constant** rather than a tunable hyperparameter.
>
> - **$k$ (Number of Salient Tokens):** Our hyperparameter analysis in **Section 6.4** provides a strong and generalizable **heuristic rule**: setting $k$ to approximately **half the prompt length** ($k \approx n/2$) yields near-optimal performance across different models and tasks. This effectively eliminates the need for an exhaustive search for $k$.
>
> - **$\omega$ (Mark Strength):** We acknowledge that $\omega$ is the only **truly sensitive** hyperparameter that requires adjustment based on the task and model. However, this level of sensitivity is standard and necessary for inference-time intervention methods to balance steering strength against distribution shift.
>
>
>
> 2. Comparison and Synergy with Chain-of-Thought (CoT)
>
> Regarding the comparison with Chain-of-Thought (CoT), we respectfully posit that directly comparing MSP to CoT as competing baselines is not an apples-to-apples comparison. CoT functions by adding informational steps to the prompt, whereas MSP is a decoding strategy designed to mitigate Attention Dilution—a phenomenon where models fail to attend to existing information.
>
> In **Section 2 of our revised manuscript**, we demonstrate that **attention dilution persists even within CoT-optimized prompts**. As the reasoning chain grows longer, the model may still lose focus on the initial constraints. Therefore, MSP is **orthogonal** to prompt optimization techniques; it is designed to enhance the model's adherence to *any* input, including CoT prompts.
>
> To empirically validate this synergy, we conducted additional experiments on **LLaMA-3.1-8B** across three benchmarks (HumanEval, MBPP, and TruthfulQA). We compared four settings: (1) Original Prompt, (2) CoT-Optimized Prompt, (3) Original Prompt + MSP (Pure MSP), and (4) CoT-Optimized Prompt + MSP.
>
> **Table: Impact of MSP on Original and CoT Prompts (LLaMA-3.1-8B)**
>
> | **Dataset**    | **Metric** | **(1) Original Prompt** | **(2) CoT-Optimized** | **(3) Original + MSP (Pure MSP)** | **(4) CoT + MSP** |
> | -------------- | ---------- | ----------------------- | --------------------- | --------------------------------- | ----------------- |
> | **HumanEval**  | Pass@1     | 31.7                    | 32.9                  | 43.9                              | 44.5              |
> | **MBPP**       | Pass@1     | 44.1                    | 45.8                  | 46.9                              | 48.1              |
> | **TruthfulQA** | Truth/Info | 78.9                    | 90.2                  | 88.4                              | 91.7              |
>
> *Note: Values for CoT and CoT+MSP are from our additional experiments; Original and Pure MSP values match Table 1 of the main paper.*
>
> **Key Observations:**
>
> 1. **MSP vs. CoT:** On code generation tasks (HumanEval, MBPP), **Pure MSP (3) outperforms CoT (2)**. This suggests that for tasks requiring strict adherence to syntactic constraints, attention steering is often more effective than simply adding reasoning steps. Conversely, on TruthfulQA, CoT outperforms Pure MSP, likely due to the reasoning depth required.
> 2. **Orthogonal Enhancement:** Crucially, regardless of whether the base input is the Original Prompt or a CoT-Optimized Prompt, applying **MSP consistently improves performance** (comparing Col 1 vs. 3, and Col 2 vs. 4).
>
> This confirms that MSP is a robust enhancement layer that can be applied *on top* of prompt engineering techniques like CoT to further unlock the model's potential.

---

> ### Author Response · Authors · 2025-11-22
> **Response to Reviewer XiCK(4/4)**
>
> ## Weakness 4 & 5
>
> We thank the reviewer for the rigorous and helpful feedback.
>
> 1、On Notation:  We have revised Eq. (3) and the related text to define $m \in \{0,1\}^n$, ensuring consistency with the input length $n$. We have also bolded the vector $\mathbf{x}$ in Eq. (2) and throughout the paper to distinguish it from scalars.
>
> 2、On Attention Claim: We agree that the statement 'assign equal attention' was imprecise. We have rewritten this sentence in Section 3 to accurately describe the phenomenon of 'attention dilution', where the attention weights on salient tokens diminish over long generations, rather than becoming mathematically equal."
>
> All modifications have been highlighted in red in the revised manuscript.

---

> > ### Comment · Reviewer_XiCK · 2025-11-27
> >
> > Thank you to the authors for the rebuttal. The new experiments helped the overall paper, and for that reason, I am raising my score. Still, some issues remain in the paper and require further evaluations and investigations:
> >
> > **Clarification on the Unintended Consequences of Masking:** Thank you for the clarification. Still, there is an issue. After computing the influence, it is used to amplify attention to specific instructions. However, such amplification can introduce unintended consequences. Although you increase attention to salient tokens, this may cause problems during generation. In the extreme case of boosting with a very high w, the model can produce incorrect outputs because it cannot attend to other contextual tokens. For this reason, there is a need for guardrails to ensure that no unintended consequences occur. Simply relying on tuning the value of w to address this issue can be computationally expensive, as acknowledged by the authors. Therefore, alternative approaches to circumvent such limitations are necessary.
> >
> > **User intent:** There are still concerns about this. For example, how do we ensure that a user’s intent about JSON formatting will be identified as the “salient” tokens by the method? It is possible that the saliency mechanism will capture other important parts of the prompt rather than the user’s intended objective. Therefore, failure cases and potential ways to mitigate them are important.

---

> > > ### Author Response · Authors · 2025-11-28
> > > **Response to Reviewer XiCK’s Follow-up Questions 1**
> > >
> > > We appreciate the reviewer's concern regarding the potential for over-steering and the need for guardrails. We are pleased to clarify that our code implementation **already includes specific engineering mechanisms** designed to prevent the "unintended consequences" of overriding context.
> > >
> > > Our implementation code (provided in the supplementary material) employs a **Probability-Modulated Steering** mechanism that acts as an adaptive guardrail.
> > >
> > > **1.Adaptive Guardrail: Probability Modulation**
> > >
> > > In our LogitsProcessor implementation, we do not simply add the influence vector $\omega \cdot v_{influence}$ to the logits. Instead, the intervention is dynamically scaled by the model's original confidence in the token.
> > >
> > > The adjustment is computed as:
> > >
> > > $$\Delta = (\omega - 1) \cdot P(t|x_{full}) \cdot (F^{original} - F^{masked})$$
> > >
> > > where $P(t|x_{full})$ is the probability of the token given the full, unmasked context.
> > >
> > > - **How this solves the issue:** If the model (attending to the full context) assigns a near-zero probability to a token (e.g., because it is syntactically incorrect or contextually irrelevant), the term $P(t|x_{full}) \approx 0$. Consequently, the steering adjustment $\Delta$ becomes negligible, regardless of the magnitude of $\omega$.
> > > - **Result:** The model ignores the Mark Text steering *if and only if* following it would contradict the global context. This effectively prevents the model from "hallucinating" or breaking syntax even under high amplification strengths.
> > >
> > > **2.Numerical Stability Guardrails (Clamping)**
> > >
> > > Furthermore, our implementation includes explicit tensor clamping to ensure stability:
> > >
> > > - We clamp the adjustment vector to the range $[-100, 100]$ to prevent "exploding logits" that could collapse the distribution.
> > > - We clamp the final logits to $[-1000, 1000]$ to ensuring numerical safety during Softmax calculation.
> > >
> > > **Conclusion:** These engineering safeguards explain the empirical robustness observed in Figure 3. The reason the model performs stably across a wide range of $\omega$ (and does not collapse at higher values) is that Probability Modulation automatically dampens the steering signal whenever it conflicts with the broader context. Therefore, MSP does not require expensive fine-tuning to be safe; it is safe by design.
> > >
> > > We will update Appendix A.5 in the final version to explicitly describe this modulation mechanism, as it addresses the safety concern practically.

---

> > > ### Author Response · Authors · 2025-11-28
> > > **Response to Reviewer XiCK’s Follow-up Questions 2**
> > >
> > > We thank the reviewer for raising this insightful point regarding user-intent alignment.
> > >
> > > First, we would like to clarify that the objective of GGMS is to identify the set of tokens that **most influence the model’s output distribution**. In the vast majority of cases, these influential tokens naturally align with, or are strongly correlated with, the user's intent. However, as the reviewer correctly points out, we cannot guarantee perfect alignment in *all* scenarios—particularly in cases where:
> > >
> > > 1. the prompt contains multiple semantically important segments, leading to a more diffuse salience pattern;
> > > 2. the user intent (e.g., formatting requirements such as JSON) is expressed only weakly within the prompt;
> > > 3. the base model itself exhibits low sensitivity to certain constraint tokens, resulting in low gradient magnitude for those tokens.
> > >
> > > In such situations, MSP may not automatically include certain intent-bearing tokens into the salient set.
> > >
> > > Nevertheless, we emphasize that **MSP is fully compatible with lightweight, training-free mitigation strategies**. Because MSP is a plug-and-play framework, the selected salient token set **$\mathcal{M}$ is inherently interpretable, inspectable, and editable**. For applications with strict formatting or structural constraints, users can easily review the GGMS-selected tokens and, when necessary:
> > >
> > > > **manually force-include specific intent tokens (e.g., “JSON”, “format”, “{ }”, etc.) into the $\mathcal{M}$ pool.**
> > >
> > > This optional step is simple from an engineering standpoint and does not compromise the generality or efficiency of our method. We have added a discussion of this flexibility as a practical mitigation mechanism.

---

### Official Review · Reviewer_ePPC · 2025-11-01

**Soundness:** 2
**Presentation:** 3
**Contribution:** 2
**Rating:** 4
**Confidence:** 4

**Summary:**

In this paper, the authors propose a method to reduce the attention dilution problem in LLMs. First, the method picks out the salient tokens in the input text by using gradient optimization to maximize the probability of change of output tokens. Then, during the decoding stage, the mask and unmasked input are fed to the model to get logit vectors. The final vector used is the weighted sum of two vectors. Experiments on five datasets and three LLMs demonstrate the effectiveness of the proposed method over baseline and two SOTA models.

**Strengths:**

1. Attention dilution is an interesting and important issue in LLMs and needs more research on it.
2. The method is intuitive to increase the attention of the model towards certain tokens in the source text.
3. Experiments demonstrate the significant improvement of the proposed method over the baseline.
4. The writing of the method section is relatively clear and easy to follow.

**Weaknesses:**

1. The research question is unclear. Although the abstract and introduction mention prompt optimization, the input in the experiments is still native input, where no prompt optimization methods are applied. This makes the core problem, attention dilution, suspicious and unclear. Are you focusing on the attention problem of the original prompt or the optimized prompt, or both? If you focus on the original prompt, does attention dilution exist (according to the abstract)? Since the instructions of the proposed datasets are short, is there any analysis to show that the model really suffers from this problem?

2. Lacks some baselines. Although SOTA and vanilla models are included, it is still unclear whether the mask generation algorithm is SOTA. For instance, can we directly ask GPT what the most salient tokens are in the sentences? Or do we have a metric to evaluate the quality of the mask? In addition, some token compression techniques might also serve as the baselines, such as LongLLMLingua.

3. Lacks some analysis. First, it is not certain how the proposed method improves the performance. Does the proposed method let the model notice some important tokens so that the model can answer the question correctly? At least some case studies should be helpful. Second, an analysis of attention dilution (such as an attention map) might enhance the understanding of the core issue that the paper wants to solve. Third, since some of the tokens in the input are directly masked, thus, the influence of grammatical correctness may lead to a response change, although the token itself might not be semantically important. For instance, sometimes LLM cannot generate y because the sentence after the mask is a wrong sentence.

**Questions:**

See weakness.

---

> ### Author Response · Authors · 2025-11-22
> **Response to Reviewer ePPC(1/3)**
>
> ## Weakness 1
>
> We thank the reviewer for this insightful comment. We apologize for not making the scope of ``attention dilution'' sufficiently clear in the original submission.
>
> First, we clarify that attention dilution is a fundamental phenomenon that occurs during the autoregressive generation of LLMs, regardless of whether the prompt is original or produced by a prompt optimization method. Our work therefore addresses both situations. MSP is orthogonal to existing optimization techniques and is designed to mitigate a core limitation of LLM decoding rather than to replace prompt optimization.
>
> To address the reviewer's concern regarding the relatively short instructions in the evaluated datasets, we conducted an additional analysis in the revised version. Following the prior work, we measured the percentage of attention allocated to the user prompt during autoregressive decoding. Specifically, we evaluated three widely used open-source LLMs on all 164 HumanEval tasks under two settings: (1) the native prompt and (2) the prompt augmented with Chain-of-Thought (CoT). For both settings, we employed a gradient-based attribution method to quantify how much the model attends to the prompt as generation progresses.
>
> As shown in Figure 1 (revised version), all models exhibit a clear and consistent decline in attention to the prompt as more tokens are generated, under both native and CoT-optimized prompts. This empirically verifies that attention dilution is not only present but can persist even when stronger prompting strategies are applied. Detailed computational procedures are provided in the Appendix.A.1 of the revised version.
>
> These findings demonstrate that the problem addressed by MSP is real, general, and not limited to a particular type of prompt, which further justifies the motivation and scope of our method.

---

> ### Author Response · Authors · 2025-11-22
> **Response to Reviewer ePPC(2/3)**
>
> ## Weakness2
>
> We thank the reviewer for raising an important point regarding baselines and mask-quality evaluation.
>
> We first clarify that the objective of GGMS differs fundamentally from both token compression methods (e.g., LongLLMLingua) and GPT-based salience identification. Methods such as LongLLMLingua rely on a small proxy model to determine which tokens are “important.’’ However, the tokens identified as important by a small proxy model are not necessarily the ones that the target LLM truly relies on. Strictly speaking, this discrepancy renders them unsuitable as direct baselines; nevertheless, we include them to ensure a comprehensive comparison. Conversely, GPT-based salience relies solely on the model's explicit assessment without accessing gradients or internal signals, which—as confirmed by our experiments—often leads to highly unstable and erroneous attributions.
>
> To quantitatively evaluate the quality of the masks produced by GGMS, we follow the evaluation protocol of Cohen-Wang et al. (2024) [1]and adopt two widely used metrics:
>  (1) **Semantic Similarity (CodeBERT Sim)**, which measures the semantic distance between the original and masked prompts (lower is better), and
>  (2) **Top-k Log-Probability Drop(LDP)**, which measures how much the target output probability decreases when the selected tokens are ablated (higher is better).
>  These metrics directly assess whether the selected tokens are indeed the ones most influential to the target model.
>
> We further include four baselines for mask generation: Random, Genetic Algorithm (GA), LongLLMLingua[2], and GPT-4 Attribution. As shown in the table below, the Genetic Algorithm (GA) yields strong attribution quality but incurs prohibitive computational cost—on Qwen-32B it requires over 16 minutes per sample on average, making it impractical for real-world use. In contrast, LongLLMLingua and GPT-4 Attribution are computationally efficient but perform poorly on both attribution metrics. Notably, GPT-4 Attribution even produces reversed attributions in some settings (e.g., a negative Drop score of –0.01 on LLaMA-3.1-8B), indicating high instability and suggesting that it may identify entirely irrelevant or noisy tokens.
>
> In comparison, GGMS achieves the best trade-off between attribution quality and computational efficiency, demonstrating both its effectiveness and practicality. We have added the full results and discussion to Section 6.3.1 of the revised manuscript (highlighted in red).
>
> |                   |          | **LLaMA-3.1-8B** |          |          | **CodeLlama-13B** |          |          | **Qwen-3-32B** |          |
> | :---------------: | :------: | :--------------: | :------: | :------: | :---------------: | :------: | :------: | :------------: | :------: |
> |    **Method**     |  Sim ↓   |       LDP↑       |   Time   |  Sim ↓   |       LDP↑        |   Time   |  Sim ↓   |      LDP↑      |   Time   |
> |      Random       |   0.82   |       1.00       | **0.00** |   0.87   |       0.13        | **0.00** |   0.77   |     10.11      | **0.00** |
> | Genetic Algorithm | **0.53** |     **5.88**     |   2.80   | **0.49** |       1.28        |   6.68   | **0.65** |      9.96      |  16.94   |
> |   LongLLMLingua   |   0.91   |       1.01       |  *0.01*  |   0.89   |       1.30        |  *0.01*  |   0.81   |      7.10      |  *0.01*  |
> | GPT-4 Attribution |   0.87   |      -0.01       |   0.03   |   0.84   |      *1.47*       |   0.07   |   0.78   |      6.09      |   0.03   |
> |  **GGMS (ours)**  |  *0.77*  |      *3.34*      |   0.10   |  *0.80*  |     **1.75**      |   0.45   |  *0.71*  |   **14.35**    |   1.05   |
>
> [1]ContextCite: Attributing Model Generation to Context(NeurIPS'24)
>
> [2]LongLLMLingua: Accelerating and Enhancing LLMs in Long Context Scenarios via Prompt Compression (ACL'24)

---

> ### Author Response · Authors · 2025-11-22
> **Response to Reviewer ePPC(3/3)**
>
> ## Weakness3
>
> We thank the reviewer for the thoughtful analysis suggestions. We address the three points below.
>
> **1. Case studies illustrating how MSP helps the model focus on important tokens.**
> We have added qualitative analyses in Appendix A.7 of our revised manuscript, including step-by-step examples showing how MSP identifies salient prompt tokens and amplifies their influence during decoding. These case studies demonstrate that MSP indeed enables the model to better attend to the crucial information required for generating correct answers.
>
> **2. Empirical analysis of attention dilution.**
>  To strengthen the understanding of the core issue our method aims to solve, we added an empirical study in Section 2(revised manuscript) that quantifies attention dilution across multiple LLMs. Consistent with prior  work, we observe a clear decline in the model’s attention to the user prompt as generation progresses, confirming that attention dilution is a real and general phenomenon.
>
> **3. Clarifying a major misunderstanding about masking.**
>  The reviewer expressed concern that masking tokens might result in ungrammatical inputs, potentially altering model behavior for reasons unrelated to semantic importance. This concern stems from a misunderstanding of our method. We emphasize that **the model never generates from masked inputs**.
>
> During decoding step $i$:
>
> - The model always computes its actual output distribution from the **full, unmasked input** $E_i$, producing logits $F_i^{\text{original}}$.
>
> - We additionally perform a **parallel forward pass** using a masked version $E_i^{\text{masked}}$ *only* to compute the difference in logits:
>
>   $v_{\text{influence}}=F_i^{original}−F_i^{masked}.$
>
>   This vector captures the contribution of the salient token set.
>
> - As shown in Eqs. (5)–(8), the final decoding logits are:
>
>   $F_i^{augmented}=F_i^{original}+ω v_{influence},$
>
>   meaning that the model’s generation is **always based on the original, unmasked input** $E_i$, with $v_{\text{influence}}$ providing an additional enhancement signal.
>
> Therefore, no masked or ungrammatical input is ever provided to the model during generation. The masking mechanism is **only used to estimate token influence**, not to construct the input used to produce the final response.

---

### Note · Authors · 2026-01-29

I have read and agree with the venue's withdrawal policy on behalf of myself and my co-authors.

---

### Meta-Review · Area_Chair_SFgA · 2026-01-06

**Summary:**

The submission targets “attention dilution” in LLM decoding. It selects salient input tokens via a gradient-guided mask search, then steers generation by combining logits from masked and unmasked passes, effectively amplifying contributions from the selected tokens. Experiments on several datasets and models show gains over baselines. Reviewers appreciate the motivation, clarity, and breadth of evaluation, and note substantial improvements on some tasks. However, core concerns remain. The problem framing and scope are unclear with respect to prompt optimization versus intrinsic decoding issues; evidence that attention dilution is the central failure mode across evaluated settings is limited. Baselines for both steps are incomplete or imperfectly matched, and the evaluation lacks transparent component-wise comparisons against strong alternatives including automatic span selection (e.g., AutoPASTA) and span emphasis methods (e.g., context-faithful prompting). The saliency step relies on a continuous relaxation for a discrete mask, and though practical, its faithfulness to true token importance and user intent is not decisively established; qualitative analyses help but do not fully resolve this. The method introduces nontrivial compute and several sensitive hyperparameters. Despite rebuttal additions, the positioning versus closely related work remains partial: AutoPASTA explicitly automates span selection with attention steering at inference time, and context-faithful prompting shows that careful, training-free prompting (e.g., opinion-based, counterfactual demos) can improve contextual faithfulness without steering internals.

**Reviewer Concerns:**

Addressed: The authors added analyses suggesting attention diminishes over generation, case studies of salient tokens, clarifications that masked inputs are not used for generation, and engineering safeguards (probability-modulated steering and clamping) to limit over-steering. They expanded ablations: alternative span selectors and alternative steering mechanisms, and reported runtime/quality trade-offs, showing their approach competitive versus genetic algorithms and stronger than prompt compression and GPT-based attribution in their setup. They corrected notation and softened claims about equal attention.

Outstanding: The central research question remains under-validated. It is still uncertain whether attention dilution is the dominant cause of errors in the evaluated tasks and whether steering yields robust gains across harder, diverse settings. The comparison to AutoPASTA is indirect; AutoPASTA uses automated span selection and explicit attention steering at inference without training, closely overlapping the submission’s contributions, but a direct, controlled head-to-head is missing. Similarly, context-faithful prompting demonstrates that prompting alone can increase faithfulness without internal steering; the current comparisons do not fully isolate gains beyond strong prompting baselines. Sensitivity to the steering weight and per-task tuning remains a practical concern despite guardrails. Finally, alignment with user intent is not guaranteed; failure modes and mitigations are acknowledged but not systematically quantified.

**Reviewer Scores:**

Based on the discussion, one reviewer rose from 4 to 6 after new experiments; another rose modestly but retained concerns; others stayed at 2–4 citing methodological and evaluation gaps. If discussion continued, I would expect modest upward movement for reviewers who prioritized added ablations, but persistent reservations for those focused on positioning and robustness.

---

### Decision · Program_Chairs · 2026-01-26

Reject